# Towards Generic Interface for Human-Neural Network Knowledge Exchange

## Abstract

Neural Networks (NN) outperform humans in multiple domains. Yet they suffer from a lack of transparency and interpretability, which hinders intuitive and effective human interactions with them. Especially when NN makes mistakes, humans can hardly locate the reason for the error, and correcting it is even harder. While recent advances in explainable AI have substantially improved the explainability of NNs, effective knowledge exchange between humans and NNs is still under-explored. To fill this gap, we propose Human-NN-Interface (HNI), a framework using a structural representation of visual concepts as a "language" for humans and NN to communicate, interact, and exchange knowledge. Take image classification as an example, HNI visualizes the reasoning logic of a NN with class-specific Structural Concept Graphs (c-SCG), which are human-interpretable. On the other hand, humans can effectively provide feedback and guidance to the NN by modifying the c-SCG, and transferring the knowledge back to NN through HNI. We demonstrate the efficacy of HNI with image classification tasks and 3 different types of interactions: (1) Explaining the reasoning logic of NNs so humans can intuitively identify and locate errors of NN; (2) human users can correct the errors and improve NN's performance by modifying the c-SCG and distilling the knowledge back to the original NN; (3) human users can intuitively guide NN and provide a new solution for zero-shot learning.

## 1 Introduction

From medicine (Shen et al., 2017) to self-driving cars (Buczak & Guven, 2015), machine learning (ML) systems are increasingly ubiquitous and outperform humans in multiple tasks. Given the breadth and importance of ML applications, it is critical for humans to use modern machine learning (ML) models, such as neural networks (NN), safely and trustfully, and to avoid unpredictable mistakes (Zerilli et al., 2019). This requires that humans should be involved in the ML loop. Thus, an interface between human and NN is needed, where NN and human can easily understand, interact, and influence each other. Especially when NN makes mistakes, human prior and domain expertise knowledge with causal inference ability and common sense may help NN achieve better performance. However, the lack of an effective interface makes it hard for a human to locate the reason for NN's error, not to mention correct the error. There are two main challenges for the interaction between humans and NN: (1) Interpretability, for humans, i.e., how to understand the reasoning logic of NN and how to correctly locate the reason of errors. (2) Changing NN's logic and decision, once humans locate the error of NN, i.e., how to correct it and improve NN's performance (Fig. 1).

To interpret a NN, current pixel-level interpretation methods (Zhou et al., 2016; Selvaraju et al., 2017) are limited to low-level relationships. Even recent human-intuitive concept-level explanations (Ghorbani et al., 2019a; Kim et al., 2018) do not reveal the reasoning logic of NN. Visual reasoning explanation (Ge et al., 2021) mimics the reasoning logic of original NN and provide logical, easy-to-understand explanations for final decisions, but cannot directly influence NN's performance to achieve closed-loop interaction. To interact with NN, current Human-In-the-Loop ML and Interacting ML are ML-centered methods that revolve around a pipeline of re-training a model using human-curated data instead of interacting with their reasoning logic (Dellermann et al., 2021).

To solve these problems, we propose Human-NN-Interface (HNI) for knowledge exchange, with the following key contributions: (1) HNI use high-level class-specific visual concepts and their relationships to build class-specific structural concepts graph (c-SCG) for each class of interest. The

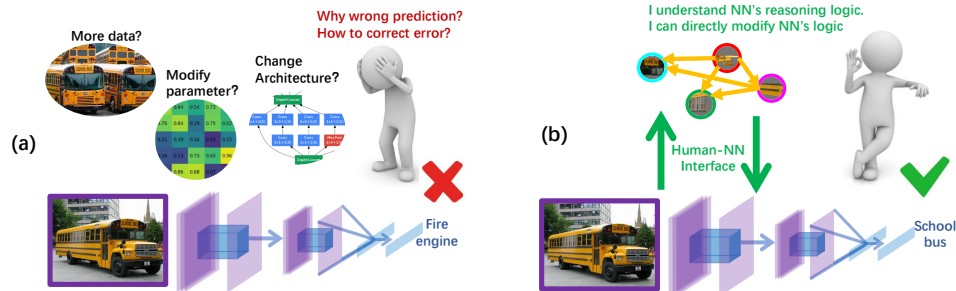

Figure 1: Human-NN Interface helps human locate and correct NN's errors by modifying NN's logic.

c-SCG is a description of the key parts (concepts; graph nodes) of an object class and their spatial relationships (graph edges). HNI allows human and NN to understand each other with c-SCG as a "language" to communicate, interact and exchange knowledge. (2) Through the NN-to-Human path, NN can use c-SCG to show their reasoning logic in a way that is intuitive to human understanding. (3) Through the Human-to-NN path, a human can analyze the reasoning logic (c-SCG) of NN and directly modify it with human prior knowledge in a very shot time (around 13s to modify each class). Then, HNI can use Graph reasoning Network and partial knowledge distillation to transfer knowledge from human back to NN, such that the NN obtains new knowledge from humans. (4) By creating new c-SCGs or modifying existing c-SCGs, humans can "teach" NN about new objects they have never seen before, which provides a new pipeline to achieve zero-shot learning.

## 2 RELATED WORK

We review research areas that share similarities with our work, to position our contributions.

**Human-AI Interaction** for Machine Learning (ML) applications aims to best combine human domain expertise and computational power of ML. To satisfy this need, Human-In-the-Loop ML (HILML) (Dellermann et al., 2021) and Interactive Machine Learning (IML) (Ware et al., 2002) have recently emerged. However, both are ML-centered methods which let humans play a "server" role around the ML process, from data production, ML modeling, to model evaluation and refinement (Maadi et al., 2021). This limits human involvement and domain expert performance. "Tell me where to look" (Li et al., 2018) uses an explainable attention map to correct segmentation errors of NN, which is thus limited to low-level relationships. User interaction was also introduced in the image generation task, Interactive Image Generation (Mittal et al., 2019) can repeatedly modify images based on modifications to the scene graph while keeping the contents generated over previous steps. In our work, we take a step toward an interface with which human users and NN can more efficiently communicate, interact and exchange knowledge between each other, with no dependency on a given list of attributes.

**Interpreting neural networks.** Research on interpretability methods for neural networks (NNs) has received more and more attention. Some try to explain NN by visualizing the correlation between each pixel of the input image and the final outputs. For instance, CAM (Zhou et al., 2016) and Grad-CAM (Selvaraju et al., 2017) can generate class-specific attention maps. Differently, several recent works focus on more human-intuitive concept-level explanations. ACE and TCAV (Ghorbani et al., 2019a; Kim et al., 2018) proposed algorithms to extract meaningful concepts from images and then produce an understandable explanation. VRX (Ge et al., 2021) uses visual concepts to explain an NN's reasoning logic. Based on this work, our HNI goes beyond this and we treat logic explanation as a foundation, allowing humans to directly modify NN's logic. Then HNI can transfer human's knowledge back to the original NN to achieve a closed-loop interaction between humans and NN.

**Graph Neural Networks** (GNN) are deep learning methods that operate on graph domains, which learn to represent graph nodes, edges, or subgraphs by low-dimensional vectors (Zhou et al., 2021). Considering the trackable information-communication properties of GNNs, many reasoning tasks have adopted GNNs as a tool, such as VQA (Teney et al., 2017; Norcliffe-Brown et al., 2018), scene understanding (Li et al., 2017a), and semantic explanations (Ge et al., 2021). In this work, we use a GNN-based Graph Reasoning Network (GRN) as the bridge between NNs and humans. It takes SCG as input and transfers knowledge with NN through knowledge distillation.

**Knowledge distillation** can effectively learn a small student model from a large ensembled teacher model using soft targets (Hinton et al., 2015). Knowledge distillation has been widely used in privileged learning (Lopez-Paz et al., 2015), adversarial defense (Papernot et al., 2016), learning with noisy data (Li et al., 2017b). In this work, we use knowledge distillation to transfer knowledge

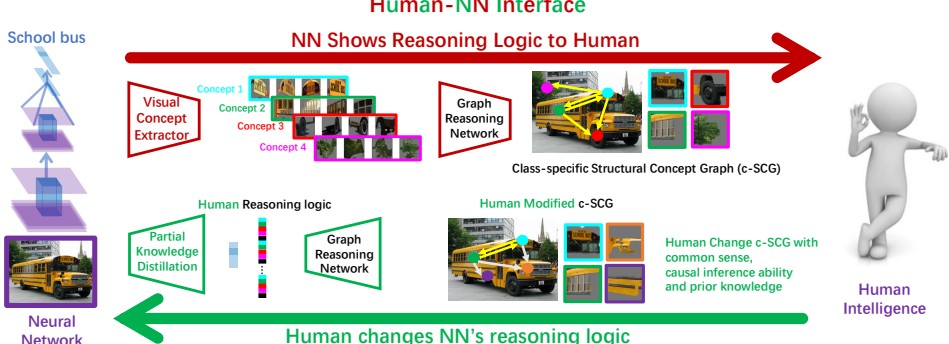

Figure 2: Pipeline for Human-AI Interface. Red arrow represents NN-to-human path, which shows the reasoning logic of original NN to human with structural concept graph (SCG). It consists of Visual Concept Extractor (discover visual concepts) and Graph reasoning Network (GRN). Green arrow represents the Human-to-NN path, which changes NN's decision making with human's prior knowledge. It consists of three steps: (1) human change SCG, (2) train GRN with human logic, and (3) transfer human knowledge to NN by partial knowledge distillation.

between a NN and a GNN-based graph reasoning network (GRN) which encodes human's prior knowledge to achieve human-NN knowledge exchange.

**Zero-shot learning** aims to train a classifier that can classify testing instances that belong to classes that are never seen before. Existing works include using the one-vs-rest solution (Verma & Rai, 2017), synthesizing pseudo instances of the unseen class (Guo et al., 2017), projecting feature space instances and semantic space prototypes into a common space (Palatucci et al., 2009), and using similar seen class as the positive instance of the unseen class (Gan et al., 2016). Huynh & Elhamifar (2020) conduct compositional zero-shot learning by using a feature composition framework to extract and combine features of attributes to construct fine-grained attributes for unseen classes. Jia et al. (2021) use active zero-shot learning which promotes human-AI teaming by actively modifying the class-attribute matrix. However such attribute labels may not be always available in real-world scenarios. Here, we propose a new pipeline with HNI, which makes no assumption on the availability of attribute label, humans help build the understanding of the new class using learned visual concepts and structure. Then HNI transfers the knowledge of unseen class back to NN.

## 3 HUMAN-NN INTERFACE

Our proposed Human-NN Interface (HNI) to bridge the interaction between human and neural networks is visually summarized in Fig. 2. There are two main path.

### 3.1 NN-TO-HUMAN

NN-to-human path explains the NN's reasoning logic for each decision (instance-level explanation, see Appendix Sec. A.1) and more importantly, the understanding of NN for each class, represented as class-specific Structural Concept Graph (c-SCG). Each c-SCG is bound to one class (Fig. 2 shows the c-SCG of school bus), where the nodes represent the important visual concepts that original NN considered most important in identifying the class of interest, and edges represent the pairwise structural relationships (dependencies) between concepts. As shown in Fig. 2 (top), given a trained NN, there are two main steps in order to explain the reasoning logic of NN to human users:

(1) Using **Visual Concept Extractor (VCE)** to discover representative visual concepts for each class of interest. The detailed procedure follows Appendix Fig. 8(a) of (Ge et al., 2021): To discover concepts for each class, we collect 50 to 100 images of the class. We first use top-down gradient attention (Grad-Cam (Selvaraju et al., 2017)) to constrain the relevant regions for concept proposals to the foreground segments, thereby ruling out irrelevant background patterns for this class. Then we follow the same workflow as the ACE paper (Ghorbani et al., 2019b): multi-resolution segmentation, feature extraction, clustering patches in latent space to obtain the concept candidates, and sorting these concept candidates based on importance score, similar to (Kim et al., 2018). After that we obtain the concept pool (each concept is represented by one mean feature vector, sorted by importance score) for each class, which will serve as a source of concept candidates when human users modify concepts (nodes) for c-SCG. To build c-SCG that reveals the reasoning logic for each class of interest of the original NN, we select the top $k$ ($k$=4 in our experiments) important concepts and their mean feature vectors as nodes. Each directed edge in an SCG edge$_{ji} = (v_j, v_i)$ has two attributes: 1) representation of the spatial structural relationship between nodes; 2) dependency $e_{ji}$ (a

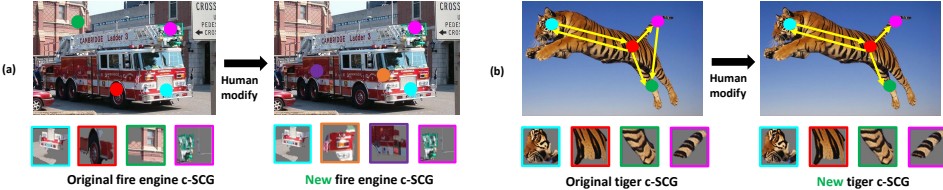

Figure 3: Examples of human modification of class-specific Structural Concept Graph (c-SCG). Each class has only one c-SCG, we use one instance to visualize the states before and after human users modifying c-SCG of the corresponding class. (a) Node (concept) modification: human user find that the NN's original concept 2 (in green, building window) to be non-causal inference and concept 3 (in red, wheel) is not discriminative and representative for this particular type of vehicle. So the human user chooses two new concepts from concept pool and replace them on the c-SCG. (b) Edge (concept relationship) modification: human user finds that the spatial relationship between concept 2 (in green, legs) and 4 (in pink, tail) to be unstable and the two concepts have minimal dependency on each other, hence deleted the edge between them on the c-SCG.

trainable scalar) between concepts $v_i$, $v_j$. The edge features can reveal the underlying causal logic of interactions between visual concepts which is crucial for the final decision. c-SCG extract such relationship between visual concepts during training and incorporate them as edge features in c-SCG. By showing c-SCG for each class of interest, NN-to-human path provides easy-to-understand insights for human users on the reasoning process of NN, which is also a foundation for the Human-to-NN path. The edges (structural relationship and dependency) are fully connected at the beginning and then, using the learning in the following step 2, we select and only keep the important edges.

(2) Using **Graph Reasoning Network (GRN)** to mimic the decision-making process of original NN with knowledge distillation. As a GNN-based network, GRN takes a graph as input. To train GRN based on the built c-SCGs (one for each of $n$ classes of interest), we need to establish connections between training images and the c-SCGs. To this end, we create, for each training image $I$, a set of up to $n$ image-level structural concept graphs (I-SCGs). Each I-SCG is computed from both the training image $I$ and one of the $n$ c-SCGs: Given the input image $I$, we use multi-resolution segmentation SLIC (Achanta et al., 2012) to break the image into patches, which become the concept candidates (similar as Fig. 4 step 1). In the concept matching step (similar as Fig. 4 step 2), for each class of interest $c$, we match features of the segmented patches to the stored anchor representation (i.e., mean feature vectors) of top $k$ concepts deemed important for $c$ using a similarity metric (e.g. Euclidean distance). When at least one of the top $k$ concepts of class $c$ is detected in image $I$, an I-SCG for class $c$ will be constructed based on the template from the c-SCG of class $c$. Here I-SCG uses patch features instead of the concept anchor features as node features and calculate the edge features based on spatial relationship between detected concepts in image $I$. This way we can have up to $n$ I-SCGs generated for the input image $I$ considering all $n$ classes of interest (More details in Appendix Sec. B.2). GRN takes the I-SCGs as input and we use knowledge distillation to transfer the decision-making logics of the original NN to GRN (similar to Appendix Fig. 8 (b) (Ge et al., 2021)). Using SCG to effectively reveal the reasoning logic of original NN has been validated with extensive experiments in (Ge et al., 2021), in which the authors evaluated the logical consistency and faithfulness between SCG explanation and original NN. (More discussion in Appendix Sec. A.2.2).

## 3.2 HUMAN-TO-NN

Human-to-NN path transfers human's knowledge to NN, in order to improve original NN's performance and generalizability. There are three main steps: (1) User modifies c-SCG: after understanding the NN's reasoning logic with NN-to-human path, users can verify whether the decision logic is reasonable or consistent with their understandings. If not, human users are able to actively correct the decision logic by updating the c-SCG (e.g., deleting a visual concept and changing the structural relationship between concepts) efficiently (about 13s for each class in our experiments). (2) To represent human-modified logic, we use the modified c-SCG as template to automatically rebuild I-SCGs for images and train a new Graph Reasoning Network (GRN), with ground truth image labels. (3) To let the original NN learn human-modified logic, we propose partial knowledge distillation to transfer the logic of GRN, which has incorporated the knowledge and prior from human users, back to the original NN. We describe the three steps in details in the following subsections.

### 3.2.1 HUMAN MODIFY C-SCG

NN's understanding of any specific class can be shown as a single c-SCG: the nodes (visual concepts) represent the crucial visual evidence or clue for NN to identify this class; the edges encode the

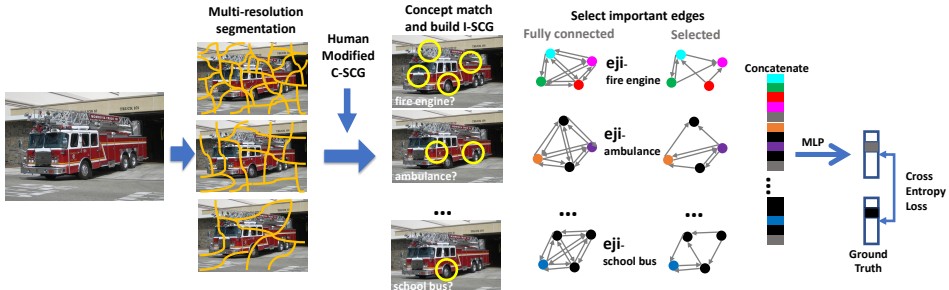

Figure 4: Pipeline of training Graph Reasoning Network with Human modified c-SCG. Given input $I$, we conduct multi-resolution segmentation and concept match step. In concept matching step, yellow circle represent the matched concepts for each class-of-interest (black dummy nodes denotes undetected concepts).(More details in Sec. 3.2.2)

structure relationships and dependency between concepts. After understanding the meaning of c-SCG, human users can then intuitively make modifications to c-SCGs (e.g., removing the incorrect nodes/edges) based on their knowledge or other priors, in order to improve the NN's performance. There are two main types of modification: nodes and edges, corresponding to changing the visual concepts and the relationships between concepts respectively.

**Node (concept) modification:** human users can easily identify non-casual concepts in c-SCG. Fig. 3(a) shows an example of node modification. In some cases, nodes may be irrelevant to the class-of-interest (e.g., a background objects always appear together with object of interest; e.g., see building window and fire truck), or not representative/unique (e.g., an object part that is common among many classes; see wheel and red dot). To substitute these two concepts with more representative and discriminative ones, human users can go back to the concept pool extracted by the VCE in the NN-to-human path and select better visual concepts to improve the c-SCG (Fig. 3(a)).

**Edge (concept relationship) modification:** Edges shown by c-SCG are the important dependencies selected based on the values of $e_{ji}$. Humans can modify them to remove non-stable or independent relationships between concepts. This may happen when substantial biases exist in training, when NN may discover stable and dependent relationships between concepts which in fact do not always hold in real-world scenarios (e.g., relative position of a tiger's leg and tail in Fig. 3(b)). Human users can simply remove this edge on the c-SCG to correct the bias. In practice, modifications of nodes and edges can happen simultaneously to handle more complex situations. Note that **to modify the decision reasoning logic for one class, human users only need to modify the corresponding c-SCG once**: e.g., they can substitute one concept with another by changing the mean vector, or add/delete edges. They do not need to modify image-level I-SCG for each training image. After updating the c-SCG, our framework automatically applies this modification to all image-level I-SCGs. Based on our human study with 11 human subjects, the average time to finish one class logic modification is 13s. Human modification of c-SCGs is the first step in the human-to-NN path, where our framework provides an intuitive way for human users convert their knowledge, common sense and priors into a description in the same language that our framework uses. Next we will show how the knowledge of human users can be transferred back to NN in the following subsections.

### 3.2.2 TRAINING GRAPH REASONING NETWORK (GRN) WITH HUMAN'S LOGIC

Typically, the set $S_I$ of classes that require human intervention is a subset of the set of all $S$ classes ($S_I \subset S$). This setting is flexible and efficient: no matter how many classes the original NN can predict (e.g., 1000 classes in ImageNet), user may only want to modify the logic of a small subset of classes in question (e.g., some vehicles are easily confused with each other). In this case, we build a GRN that targets the logics of these classes only, which is more efficient to users. For each class of interest $c \in S_I$, NN-to-human path reveals its reasoning logic c-$SCG_{NN}^c$. After human's analysis and modification, some class may have updated c-SCGs, c-$SCG_H^c$ after incorporating human user's knowledge. c-SCG as a class representation cannot produce final decisions by itsel hence we reuse the GRN from NN-to-Human path to infer a prediction.

Fig. 4 shows the pipeline of training GRN with the c-SCG updated by human users. Given an input image $I$, the first two steps of the processing(i.e. Multi-resolution segmentation and Concept match to build I-SCG) are same as the GRN training in NN-to-human path while the objective is different. Here our goal is to obtain better performance by incorporating human user knowledge. the matched I-SCGs go through the graph convolution backbone and MLP in GRN and finally predict the image

label with cross-entropy loss as the objective function. The trained GRN can then produce decisions based on human-corrected reasoning logic because the input I-SCGs are derived from c-$SCG_H^c$. (See Appendix Sec. B.5 for more implementation details).

### 3.2.3 TRANSFERRING REASONING LOGIC TO NN WITH PARTIAL KNOWLEDGE DISTILLATION

The last step to is to transfer the user-updated reasoning logic and knowledge in GRN back to the original NN. To avoid catastrophic forgetting and negative impact on the classification performance of other classes, we proposed partial knowledge distillation as knowledge transfer method.

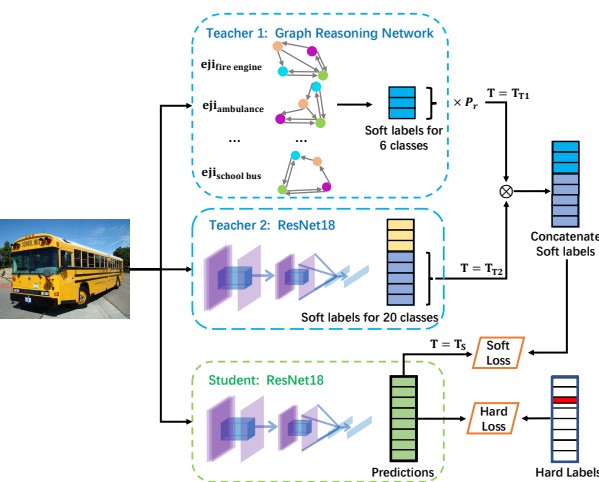

Figure 5: The pipeline of Partial Knowledge Distillation. Different from traditional knowledge distillation, partial knowledge distillation adopts two teachers with different expertise: GRN (teacher 1) focuses on class of interest (6 classes in this example), and fixed original NN (teacher 2) will focuses on the rest of classes (the classes we don't want to change, 14 classes in this example). After distillation with different temperatures and concatenation, we can use both soft labels and hard labels to train the student model.

Fig. 5 illustrates the process of partial knowledge distillation transferring human user's knowledge from GRN back to the original NN. As described in Sec. 3.2.2, modified classes $S_I$ is a subset of all class set $S$. For unmodified classes set $S_U = S \setminus S_I$, we want to maintain their reasoning logic while we update those of of $S_I$ in original NN. Hence two teacher models provide soft labels together. GRN $Net_{T1}$ provides the probabilities of modified classes, the original NN $Net_{T2}$ with *fixed* parameters provide the probabilities of unmodified classes. Student model $Net_S$ shares the same architecture of the original NN and is initialized with the weights of original NN. Formally, the overall loss during partial knowledge distillation is as follows:

$$L = \alpha L_{soft} + \beta L_{hard} \tag{1}$$

where $\alpha$ and $\beta$ are the weighting of the two terms during distillation. For the soft label term:

$$L_{soft} = -\sum_i^N \hat{p}_c^{T_T} log(q_c^{T_S}); \quad q_i^{T_S} = \frac{exp(z_i/T_S)}{\sum_k^N exp(z_k/T_S)} \tag{2}$$

where $\hat{p}_c^{T_T}$ denotes the probability value of class $c$ in the combined soft label with temperature $T_T$ from two teacher models. $q_c^{T_S}$ denotes the probability value of class $c$ in the student prediction vector with temperature $T_S$. $N = |S|$ denotes the number of classes in the original NN. For $q_c^{T_S}$, $z_c$ denotes the logits of $Net_S$, which is the unnormalised predictions. The combined soft label $\hat{p}^{T_T}$ is the combination of two soft labels $p^{T_{T1}}$ and $p^{T_{T2}}$ from two teacher models $Net_{T1}$ and $Net_{T2}$. $p^{T_{T2}}$ is a vector with length $N$, $p^{T_{T2}} \in \mathbb{R}^N$, while $p^{T_{T1}}$ is a vector with length $n$, $p^{T_{T1}} \in \mathbb{R}^n$, where $n = |S_I|$ denotes the number of classes in the GRN, which is also the number of modified classes, $v_c^1$ and $v_c^2$ denote the logits of the teacher models $Net_{T1}$ and $Net_{T2}$ respectively:

$$\hat{p}_c^{T_T} = \begin{cases} p_c^{T_{T1}} Pr, & \{c \in S_I\} \\ p_c^{T_{T2}}, & \{c \in S \setminus S_I\} \end{cases} \quad s.t. \quad p_c^{T_{T1}} = \frac{exp(v_c^1/T_{T1})}{\sum_k^n exp(v_k^1/T_{T1})} \quad p_c^{T_{T2}} = \frac{exp(v_c^2/T_{T2})}{\sum_k^N exp(v_k^2/T_{T2})} \tag{3}$$

in which $Pr \in (0, 1]$. To obtain the combined soft label $\hat{p}_c^{T_T}$, we first compute the sum of the probability of all classes of interest in $p^{T_{T2}}$. $Pr = \sum p_c^{T_{T2}}$ for all $\{c \in S_I\}$, which represents the probability proportion of the $n$ modified classes w.r.t. all classes $N$ in the original NN $Net_{T2}$. We then replace the value of class of interest in $p^{T_{T2}}$ with the scaled value in $p^{T_{T1}}$ to form the combined soft label. The prediction of teachers can be erroneous, and we use ground-truth labels as hard labels to provide stronger constraint to $Net_S$ correcting these errors from teacher models.

$$L_{hard} = -\sum_{c=1}^N g_c log(q_c^1) \tag{4}$$

Table 1: ImageNet 20 class classification performance with node modification via HNI

| Category | **Beach wagon** | **Ambu lance** | **School bus** | **Jeep** | **Fire engine** | **Recrea vehicle** | Horse | Side winder | Irish setter | Harte beest |
|---|---|---|---|---|---|---|---|---|---|---|
| Original NN Accuracy | 0.54 | 0.69 | 0.80 | 0.53 | 0.73 | 0.68 | 0.93 | 0.64 | 0.73 | 0.67 |
| Modified NN Accuracy | **0.67** | **0.75** | **0.88** | **0.65** | **0.79** | **0.79** | 0.92 | 0.66 | 0.72 | 0.70 |

| Category | Bassinet | Face powder | Green house | Manhole cover | Plane | Shield | Toilet tissue | Ram | Water buffalo | Zebra |
|---|---|---|---|---|---|---|---|---|---|---|
| Original NN Accuracy | 0.64 | 0.55 | 0.76 | 0.77 | 0.54 | 0.43 | 0.53 | 0.60 | 0.54 | 0.78 |
| Modified NN Accuracy | 0.62 | 0.58 | 0.75 | 0.79 | 0.52 | 0.41 | 0.55 | 0.59 | 0.57 | 0.80 |

where $g_c$ denotes the ground truth label for class $c$, $q_c^1$ is the probability value of class $c$ in the student prediction vector under temperature 1. Eq. 1 transfers knowledge from GRN back to the original NN (see Appendix for more implementation details).

To summarize, we use Graph Reasoning Network (GRN) in both the NN-to-Human path and Human-to-NN path respectively (Fig. 2) with the same structure. In the NN-to-Human path, GRN simulates the reasoning logic of the original NN with knowledge distillation (detailed workflow see Appendix Fig. 8 (b)(Ge et al., 2021)). In the Human-to-NN path, after human users modifying the c-SCG for classes of interest, GRN uses the modified c-SCG to automatically derive I-SCGs for each image which will be used to train the GRN using ground truth labels (Fig. 4). Once the training of GRN is done, we transfer the knowledge of GRN back to the original NN, fow which we proposed partial knowledge distillation, where the newly-trained GRN becomes the teacher model to train the original NN using GRN's outputs as soft labels (Fig. 5). We summarize the input/output details of HNI pipeline and each module/process in Appendix Table. 4.

## 4 Experiments on Human-NN Interface Applications

HNI can be used as a generic interface for knowledge exchange between human users and NN. We conduct experiments to demonstrate several applications of HNI: (1) Human users improving NN's performance by updating the logic of important concepts (Sec. 4.1.1) and relationships between them (Sec. 4.1.2), (2) Human users guiding NN in Zero-shot Learning(Sec. 4.2).

### 4.1 Human improve NN's performance with HNI

Through the Human-to-NN path (Sec. 3.2), humans can modify c-SCGs with their knowledge and then transfer the knowledge back to original NN. This process can improve NN's performance when NN can not learn generalizable and robust logic during training. This is quite common especially when training data is scarce or biased. Scarce data can cause distribution mismatch between the training and test set, preventing the NN from really 'understanding' the classes-of-interest. Bias in training data can misled the learner to focus on the "wrong" patterns irrelevant to task objectives. We will show how human users can use the human-to-NN path to improve original NN's performance.

#### 4.1.1 Nodes (concepts) modification.

We use a subset of the ILSVRC2012 (ImageNet) (Deng et al., 2009) to train an 20-class classifier with ResNet-18 (Test performance see Table 1), each category consists of 200 images for training and 200 images for testing. Because the accuracy of the first six vehicles (bold) is low, we ask human users to help improving their performance with our framework. We first use the NN-to-Human path of HNI to visualize the reasoning logic of the original NN. A human user can modify the concepts in question. Appendix Fig. 11 shows the c-SCG comparison before and after modification. We then train GRN with the updated c-SCG (Sec. 3.2.2). To transfer human logic back to the original NN, we use partial knowledge distillation (Sec. 3.2.3). The test results of modified NN is shown in Table. 1. The results suggest that by incorporating human user inputs, the original NN becomes more robust and generalizes better on the test set. We also conducted a larger experiments of node and edge modification with 120 ImageNet classes (Appendix Sec. D). While the performance on the six vehicles classes improved, the performance of other classes is almost not impacted, which shows HNI can precisely modify the reasoning logic of the target classes while preserving the reasoning

Table 2: Baseline methods accuracy on modified classes in ImageNet 20 class classification

| | Beach wagon | Ambulance | School bus | Jeep | Fire engine | Recreational vehicle |
|---|---|---|---|---|---|---|
| Original NN | 0.54 | 0.69 | 0.80 | 0.53 | 0.73 | 0.68 |
| Baseline 1 **Without** Human modifying c-SCG | 0.53 | 0.66 | 0.82 | 0.53 | 0.74 | 0.64 |
| Baseline 2: add 30 images / class | 0.55 | 0.71 | 0.81 | 0.55 | 0.73 | 0.69 |
| Ours: **With** Human modifying c-SCG | **0.67** | **0.75** | **0.88** | **0.65** | **0.79** | **0.79** |

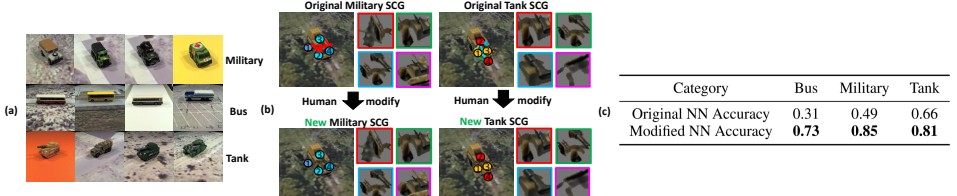

Figure 6: Example of edges (concepts relationship) modification. (a) biased iLab dataset. (b) Human user can remove incorrect edges to guide the model to ignore irrelevant concept relationships introduced by dataset bias. (c) iLab-20M three class classifier performance.

logic of others. This makes HNI more flexible and efficient (Sec. 3.2.2). (more details in Appendix). *Baseline methods.* We design the following methods as our baselines: (1) To evaluate how much human intervention boosts the NN performance, we conduct partial knowledge distillation *without* human intervention by directly supplying unmodified c-SCG to the human-to-NN path. (2) based on the time it takes for human users to modify class-specific SCG (13s/class), we asked volunteers to annotate additional images with similar effort (30 additional labeled images for each of the 5 classes) and evaluate the performance. All baseline results are shown in Table. 2 (we only compare the modified 6 vehicle class; unmodified class have similar results before and after the modification).

### 4.1.2 Edges (concepts relationship) modification.

We conduct an edges (concept relationships) modification experiment on the iLab-20M (Borji et al., 2016) dataset, which has a controllable pose to introduce bias on concept relationships. iLab-20M contains images of toy vehicles placed on a turntable using 11 cameras at different viewpoints. We tailor a subset of iLab-20M to train a three-class vehicle classifier with ResNet-18: bus, military, and tank. In the training set, each class has 400 images. We manually introduce biases in pose of each class: all buses are with pose 1, all military vehicles are with pose 2 and all tanks are with pose 3 (Fig. 6 (a)). We construct an unbiased test set where each kind of vehicle has all the 3 poses.

After training, we use the NN-to-human path to reveal the reasoning logic of original NN to humans by visualizing the c-SCG and explaining the reasoning logic on misclassified images using I-SCG (see Appendix Sec. A.1 for instance-wise explanation)). Human users found that important concepts in c-SCG are mostly in the foreground and are consistent with huamen user's own understanding of the class of interest. Explanation of incorrectly predicted images also shows similar results that most of the detected visual concepts had a positive contribution to the correct class. Thus, no node modification was needed. However, the structural relationship between concepts contributed mostly negatively, which caused incorrect predictions (Fig. 6 (b)). To eliminate the unstable/independent concept relationship, we delete all edges in SCG and train a GRN with the new c-SCG. The new GRN focus only on the existence of visual concepts during decision making, while it does not pay attention to the relationship between them. Although in general deleting all edges may be too extreme, in this case it is one reasonable way to correct the problem as most errors come from pose bias in the training set, and edge features in SCG relies heavily on the pose of target objects in this case. After the modification, we use partial knowledge distillation to modify the decision logic of NN. The final performance of the modified NN shows improvement compared with the original NN (Fig. 6(c)), demonstrating that HNI can help human to improve original NN's performance with an intuitive interface and effective mechanism in modifying the concept relationship and exchanging knowledge. (More details in Appendix). In other applications, humans may modify nodes and edges simultaneously to improve the performance of NN. To evaluate the user feedback of HNI. We conducted a human user study (in Appx. Sec. F) with responses from ML practitioners and students.

### 4.2 Zero-shot learning: Human teach NN to learn new object through HNI

Zero-shot learning (Xian et al., 2018) is a popular and challenging task, where, at test time, a learner needs to classify samples from classes not seen during training. We introduce a novel zero-shot learning pipeline with the proposed Human-NN-Interface (HNI). The high-level idea is that the understanding of a new object can be represented as class-specific c-SCG, which consists of visual concepts (nodes) and concept relationships (edges). Our HNI allow humans to design new c-SCG for new object category, with existing primitive visual concepts (nodes) or relationships (edges) discovered from other classes. The new SCG can then be distilled back to the original NN, thereby guiding the original NN to encode new object category (i.e. zero-shot learning. While learning to recognize new class, the original NN will not "forget" the old classes. In our experiment, each learned 8 objects A, B, C, D, E, F, G and H has 300 training images and 216 testing images, while the new object I, J, K has only 216 testing images. Please refer to Fig. 7 for the detailed workflow:

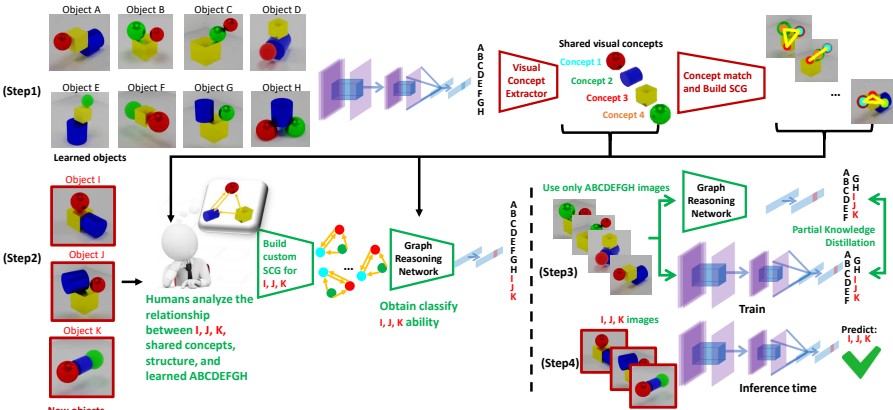

Figure 7: Zero-shot learning: Human users teach NN to learn to encode new objects with HNI. Sec. 4.2 provide explanation for each step.

Table 3: Performance of Zero-shot learning with HNI. Original NN ResNet-18 (pretrain on ImageNet) trained with images of seen objects can not identify new objects I, J, K in the test set. Human teach ResNet-18 to encode and recognize new objects I, J, K wih HNI.

| Category | A | B | C | D | E | F | G | H | I | J | K |
|---|---|---|---|---|---|---|---|---|---|---|---|
| Original NN | 0.8 | 0.79 | 0.9 | 0.74 | 0.87 | 0.78 | 0.9 | 0.9 | 0 | 0 | 0 |
| Modified NN | 0.78 | 0.78 | 0.86 | 0.72 | 0.85 | 0.8 | 0.87 | 0.89 | **0.78** | **0.9** | **0.76** |

**Step 1: NN-to-Human:** We train a classifier for 8 objects A to H. We use VCE to discover the visual concepts and find the shared concepts. We match the shared visual concepts from training images and form image-level SCGs (I-SCG) for all training images of objects A to H.

**Step 2: Human-to-NN: Building new c-SCG for new object E and train new GRN:** It is straightforward for Human to learn about a new class as they can relate the patterns and components on the new class to those they have seen in the past. We try to implement a similar mechanism here in describing the new class with SCG to GRN. We construct novel I-SCG training instances with visual concepts and relationship from learned classes, in an automated fashion. For instance, to form a I-SCG for new object I, we know some of its components are overlapping with object A to H. Hence we randomly sample one I-SCG from object A and use its concept 1 as the node of concept 1 in E's I-SCG. Similarly, we obtain I's nodes of concept 2 and 3 by randomly sampling I-SCGs of objects E and C. To construct I-SCG edges for E, we sample I-SCGs of D and form the edges of I based on their similar structures. Building I-SCG for new objects J and K are similar. We form a new I-SCG training set by adding the novel I-SCGs of I,J,K into the original training set and then train a GRN that can classify objects A to H and I, J, K.

**Step 3: Transfer knowledge from GRN back to original NN:** we use knowledge distillation to transfer the knowledge about new object E from GRN back to the original NN. In this process, we *only* use the images of A to H as training set, and only use the soft label form GRN without any hard label to avoid bias toward old classes.

**Step 4: NN learn the knowledge to encode new objects** without forgetting the knowledge about old classes. Table. 3 shows the performance of zero-shot learning with our HNI (see Appx. for more details). We made the OBJECT dataset which we created and used here public. (Appx. Sec. C).

**Comparing with mainstream zero-shot learning settings** (Geng et al. (2020); Fu et al. (2018); Xian et al. (2018)), where attribute descriptions for the images are given (e.g., stripes, horse-like shape, big four-leg animal), our method considers a more general settings and we make no assumption on the availability of attribute labels. Instead, we rely on the unsupervised mining of primitive concepts from the training dataset (without any attribute or concept labels). With different combinations of any subset of these learned primitive concepts and different structural relationships between them, we can use GCN to represent novel classes and eventually guide the NN to learn to encode them by HNI. Our method has its limitations especially when the new class can not be easily represented by the learned primitive visual concepts. However, our method is flexible in the sense that it can define new structural/spatial relationships based on learned relationships. We argue that this is an important strength of our method as it is more extensible and capable of encoding novel objects with fewer assumptions, i.e., not limited by the given list of attributes in describing relationships/structures of components/parts of novel objects.

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

APPENDIX

# A    FURTHER DISCUSSION AND EXPLORATION OF VISUAL REASONING EXPLANATION (VRX) FRAMEWORK (GE ET AL., 2021)

## A.1    INTRODUCTION

Visual Reasoning Explaination (VRX) (Ge et al., 2021), which for the first time uses GNN to mimic the reasoning logic of original NN and demonstrate the effectiveness of using Structural concept graph (SCG) to provide logical, easy-to-understand explanations for final decisions. Our NN-to-human path is based on two important module of VRX: Visual Concept Extractor (VCE) and Graph Reasoning Network (GRN). We recap some important figures (Fig. 8, Fig. 9, Fig. 10) in this section which may help understand the details of VRX framework (Ge et al., 2021).

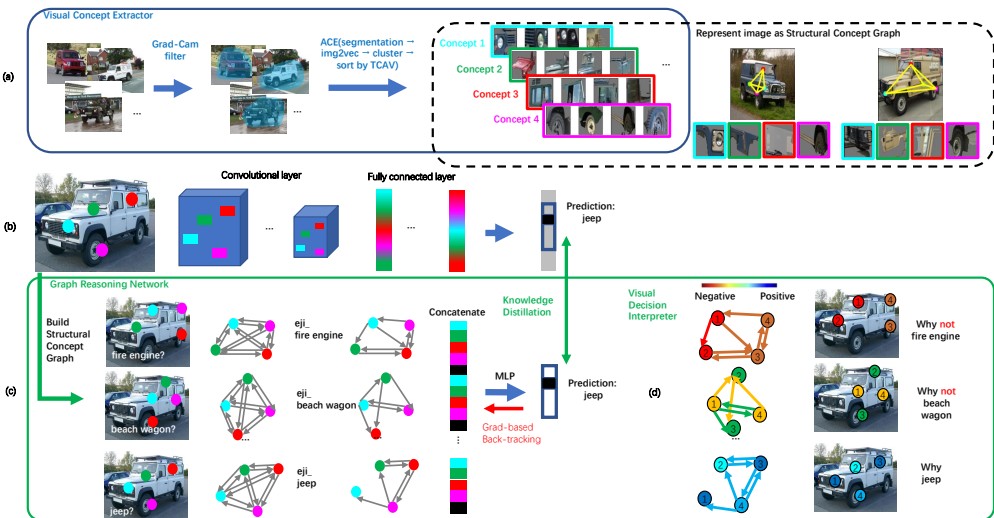

Figure 8: Image source: (Ge et al., 2021) Pipeline for Visual Reasoning Explanation framework. (a) The Visual Concept Extractor (VCE) discovers the class-specific important visual concepts. (b) In original NN, the representation of the top $N$ concepts is distributed throughout the network (colored discs and rectangles). (c) Using Visual Concept Graphs that are specific to each image class, our VRX learns the respective contributions from visual concepts and from their spatial relationships, through distillation, to explain the network's decision. (d) In this example, the concept graphs colored according to contributions from concepts and relations towards each class explain why the network decides that this input is a Jeep and not others.

For **instance-wise explanation**, with an important module Visual Decision Interpreter (VDI), VRX can meaningfully answer "why" and "why not" questions about the prediction, providing easy-to-understand insights about the reasoning process. Fig. 8 (d) and Fig. 9 give more detailed explanation.

## A.2    FURTHER DISCUSSION

### A.2.1    UNDERSTANDING OF REASONING LOGIC

In Visual Reasoning Explaination (VRX) paper (Ge et al., 2021), they define the Reasoning logic as using the structured visual concept as tools/languages to answer the question of why and why not (e.g., why the input image is an ambulance? why not fire engine?), which is a human-friendly way to understand the decision clue and logic of original NN. Thus, here we use the term "reasoning" to refer to the correlations between primitive patterns and specific classes of interest learned/encoded by the original neural network.

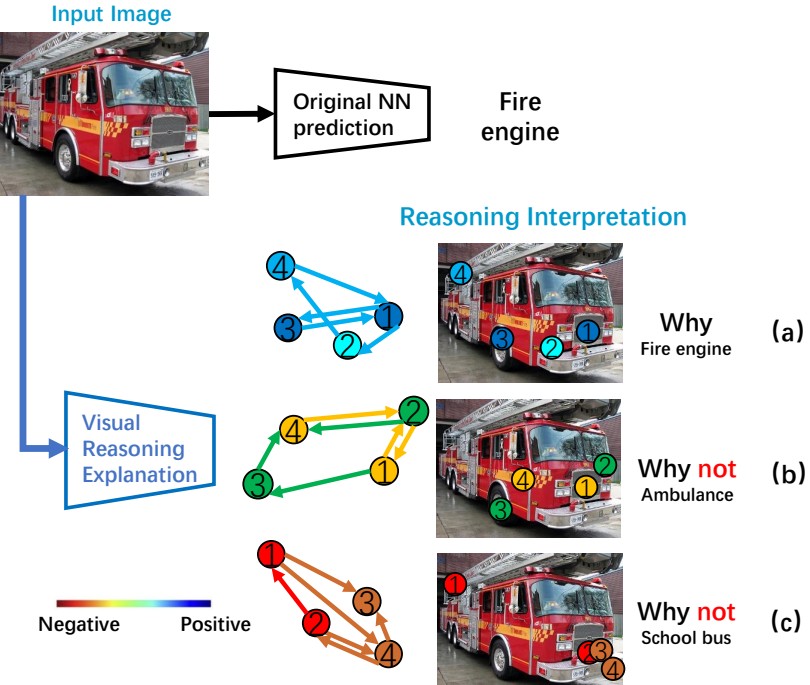

Figure 9: Image source: (Ge et al., 2021) An example result with the proposed VRX. To explain the prediction (i.e., fire engine and not alternatives like ambulance), VRX provides both visual and structural clues. Colors of visual concepts (numbered circles) and structural relationships (arrows) represent the positive or negative contribution computed by VRX to the final decision (see color scale inset). (a): The four detected concepts (1-engine grill, 2-bumper, 3-wheel, 4-ladder) and their relationships provide a positive contribution (blue) for fire engine prediction. (b, c): Unlike (a), the top 4 concepts, and their relationships, for ambulance/school bus are not well matched and contribute negatively to the decision (green/yellow/red colors).

### A.2.2 FIDELITY OF SCG EXPLANATION

In terms of the fidelity of using Structural Concept Graph (SCG) to explain the reasoning logic of original NN, here are more details: While we agree that our current pipeline does not provide a guarantee that SCG will be 100% faithful to the original NN's reasoning logic, we would like to point out a few considerations in our design:

(1) The faithfulness of our SCG to the original NN depends on several factors, including the concept selection, the number of concepts used to construct the SCG, the sampling of data used in distillation from the original NN to SCG. Since the focus of this paper is more of a feasibility study, rather than a ready-to-deploy system, each component inevitably has errors to a certain extent.

(2) We use top-down attention (e.g. Grad-CAM) to identify regions in the input image which contributed most to the prediction of each class-of-interest, which can limit the region in which we conduct concepts discovery. By doing this, we can filter irrelevant concepts and focus on those considered important in the original NN's reasoning/inference.

(3) In concept selection, we use concept activation vectors (TCAV) (Kim et al., 2018) to compute the importance score of each concept w.r.t. the prediction of a specific class-of-interest. With sufficient sampling of data, this gradient-based method can faithfully reflect the overall correlation between each visual concept and each class-of-interest in the original NN.

(4) Knowledge distillation is widely used in transferring knowledge (encapsulating the reasoning logics) between neural networks. In other words, it is transferring the input-output mapping from one neural network to another. In our framework, we adapted knowledge distillation to imitate such

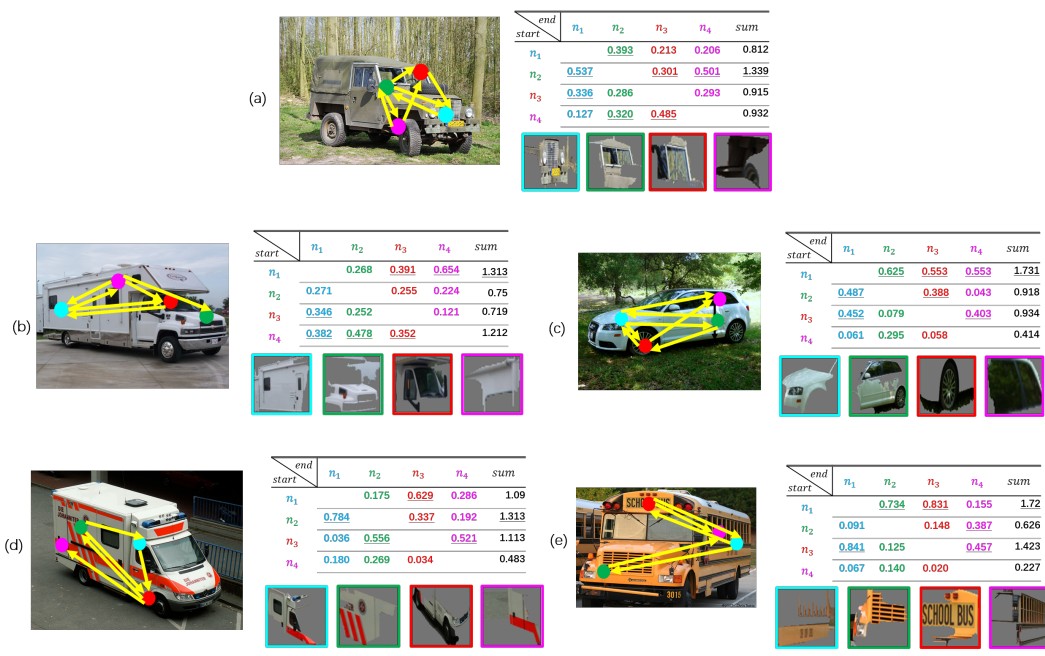

Figure 10: Image source: (Ge et al., 2021) Class-specific importance weights $e_{ji}$ highlight the important concept relationships for different classes, $e_{ji}$ (shown as tables for each class) reveals the information transformation between concepts, which shows the dependency between concepts: concept 1 and 2 contribute most information to other concepts, which makes them the 2 most discriminating concepts for a fire engine.

mapping (instead of mapping from the input image to output logits, we learn the mapping between the visual concepts and the output logits). We also add perturbation during distillation by randomly masking out one of the detected visual concepts in the input image while making sure GRN and original NN produce the same outputs. Given enough training data, GRN is expected to faithfully mimic the original NN and reveal the reasoning logics.

(5) Experimental validation. The faithfulness of using Structural Concept Graph (SCG) to explain the original NN is one key contribution of Visual Reasoning Explanation (VRX) ref [2], and in the paper, the authors conducted several experiments to prove the fidelity of SCG explanation, such as logical consistency between the explanation using SCG and the original NN. To briefly recap:

1) In Sec.4.2 of VRX paper (Ge et al., 2021), they qualitatively and quantitatively verified the logical consistency between explanation using SCG and original NN by showing that VRX can correctly locate the reason that causes the original model's wrong prediction and successfully correct the errors with the guidance of the explanation (Table 1 of VRX paper (Ge et al., 2021) shows 114/119 errors were successfully corrected). The logical consistency verification experiment shows that the explanation with SCG is faithful to the original NN.

2) Explanation sensitivity of visual and structural perturbations. The authors also conducted two experiments with the control variate method to verify the explanations of VRX are faithfully and sensitive to visual perturbation and structural perturbation respectively (Sec.4.3 and Figure 7 of VRX paper (Ge et al., 2021) ).

Based on the above-mentioned designs and experimental validations on the fidelity of SCG explanations, we argue that the explanation with SCG can be faithful to the original NN, in ideal scenarios (assuming every component's error is minimized). As part of future work, we plan to investigate the performance influence of the non-perfect fidelity of SCG. However with our current pipeline, the users can rely on the knowledge distillation loss, which can serve as a good indication of the faithfulness of the SCG to the original NN, to gain a better understanding of whether the SCG needs further improvement (e.g. with more data used in knowledge distillation, more visual concepts should be selected, etc.)

# B  IMPLEMENTATION DETAIL

## B.1  PIPELINE SUMMARIZATION

Table. 4 is the input and output summarization of the whole Human-NN Interface pipeline and each module/process.

Table 4: Input and output summarization of the whole HNI pipeline and each module/process

| Module / Process/ Pipeline | Input / collaborator | Output |
|---|---|---|
| Human NN Interface (HNI) Pipeline | Original NN | Modified NN |
| 1 NN-to-Human path | Original NN | c-SCG, Reasoning logic explanation |
| 1.1 Visual Concept Extractor (VCE) | Original NN, 50 to 100 images for each class | Important visual concepts for each class |
| 1.2 Image-level SCG (I-SCG) building | Original NN, images, visual concepts | I-SCG |
| 1.3 Graph Reasoning Network (GRN) | Original NN, Image-level SCG (I-SCG) | GRN (mimic original NN), c-SCG |
| 2 Human-to-NN path | Original NN, class-specific SCG (c-SCG) | Modified NN |
| 2.1 Human involved logic modification | c-SCGs | human-modified c-SCG |
| 2.2 GRN independent training | I-SCG built with human-modified c-SCG | GRN with human reasoning logic |
| 2.3 Partial knowledge distillation | Original NN ($Net_S$), GRN ($Net_{T1}$), Original NN (fix)($Net_{T2}$), training images | Modified NN |

## B.2  CONCEPT MATCHING

In multi-resolution segmentation step, to extract features for each patch resulting from image multi-resolution segmentation, we resize the patch and use the original NN to compute features after a specific layer, e.g. "layer4.1.conv2" layer of ResNet-18. For each discovered concept, we store the mean vector of all patches belonging to this concept as an anchor for future concept matching given any image.

In the concept match step (same as Fig. 4 step 2), for each class of interest $c$, we match candidate features to the stored anchors (mean concept vectors) of top $k$ concepts in the concept pool, and we construct an I-SCG for image $I$ and class $c$, based on similarity (Euclidean distance) between image patch feature and concept anchor feature. Specifically, if the nearest image patch regards the euclidean distance between the concept anchor feature and the image patch feature is smaller than a threshold $t$, then we will identify this patch as a detected concept. Otherwise, we will use dummy nodes (all feature values equal to a small constant $\epsilon$) to represent that undetected concept.

We empirically choose the threshold $t$ from observation when matching concepts mean vector to patches segmented from images and form image-level SCG (I-SCG). Specifically, the distance in the positive match and negative match have different orders of magnitudes in latent space, and the performance is not sensitive to the selection of $t$. We plan to explore more in this direction as to future work.

## B.3  NODE (CONCEPT) MODIFICATION

Fig. 11(a) visualize the class-specific SCG (c-SCG) of three example classes of interest. A human can modify the anti-causality and anti-common sense concepts by new concepts extracted by VCE. The new c-SCG, which merged human's reasoning logic, are shown in Fig. 11(b).

## B.4  PARTIAL KNOWLEDGE DISTILLATION

For the all partial knowledge distillation experiments in the Human-to-NN path, we use SGD with initial learning rate = 0.1, and it will multiply 0.1 every 30 epoch. Batch size = 4, the coefficient of $L_{soft}$, $\alpha = 2.5$, the coefficient of $L_{hard}$, $\beta = 1$. For Nodes (concepts) modification experiment (4.1.1 in the main paper), we use $T_s = 1, T_{T1} = 1.5, T_{T2} = 1$, For edges (concepts relationship) modification (4.1.2 in the main paper), we use $T_s = 2, T_{T1} = 2$. For zero-shot learning experiments (4.2 in the main paper), we use $T_s = 1.5, T_{T1} = 1.5$. We use knowledge distillation to train original model for 100 epochs.

For the performance influence on unmodified classes in Table. 1. We agree that if we cannot find the best partial knowledge distillation temperature hyperparameter, a slight performance fluctuation of

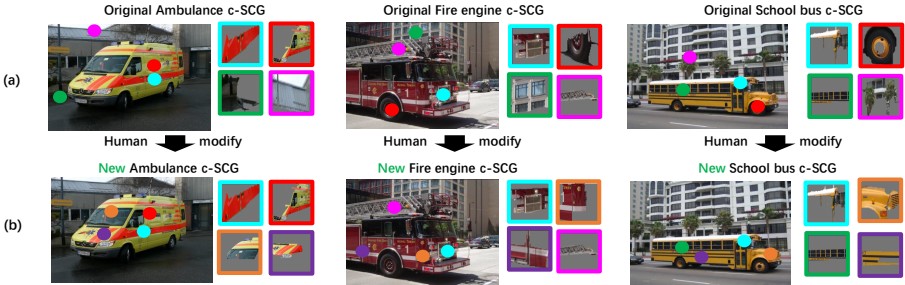

Figure 11: Results of node (concepts) modification: for the ambulance structural concept graph (c-SCG), human find NN's original concept 3 (in green frame, ground) and concept 4 (in pink frame, building roof) is anti-causal inference, due to the fact that many ambulance training images were in city backgrounds, so we choose two new concepts, side of the car (in orange frame) and car headlight (in purple frame) from discovered concept pool and substitute them. The rest examples are similar.

unmodified classes in Table. 1 may occur. We call it fluctuation because we also find some unmodified classes slightly improved after distillation. We will conduct more experiments on this direction.

### B.5 GRAPH REASONING NETWORK

**Network Structure** The network architectures of Graph Reasoning Network ($GRN$) are shown in Table. 5. $GRN$ consists of two parts, Graph Neural Network $G$ and Embedding Network $E$. $G$ takes $n$ hypotheses $\mathbf{h} = \{h_1, h_2, ...h_n\}$, (each hypothesis $h_i$ is in the form of Structural Concept Graph (SCG), and each SCG has $m_n$ nodes as well as $m_e$ edges) as input, and output $n$ feature vectors ($G(h_i)$) which concatenate all updated node and edge features of $h_i$. In $G$, we use class-specific $e_{ji}^c$ for different hypotheses in each GraphConv layer. $E$ concatenates all $n$ feature vectors from all the hypotheses into a long vector and maps it into $n$ dimensional vector ($1 \times n$) with a 4 layer MLP, where $n$ is the number of classes of interest. "node" denotes node feature, "edge" denotes edge feature, "GraphConv" is graph convolutional layer, "ReLU" denotes ReLU activation function, "BN" denotes batch normalization, and "FC" denotes fully connected layer.

| Part | Input $\rightarrow$ Output Shape | Layer Information |
|------|-------------------------------|-------------------|
| $G$ | node:($l \rightarrow 64$); edge:($4 \rightarrow 5$) | GraphConv-($e_{ji}^c$), ReLU, BN |
| | node:($64 \rightarrow 32$); edge:($5 \rightarrow 5$) | GraphConv-($e_{ji}^c$), ReLU, BN |
| | node:($32 \rightarrow 32$); edge:($5 \rightarrow 5$) | GraphConv-($e_{ji}^c$), ReLU, BN |
| | node:($32 \rightarrow 32$); edge:($5 \rightarrow 5$) | GraphConv-($e_{ji}^c$), ReLU, BN |
| | node:($32 \rightarrow 32$); edge:($5 \rightarrow 5$) | GraphConv-($e_{ji}^c$), ReLU, BN |
| $E$ | $((32 \times m_n + 5 \times m_e) \times n) \rightarrow (128)$ | FC-($(32 \times m_n + 5 \times m_e) \times n, 128$) |
| | $(128) \rightarrow (64)$ | FC-(128, 64) |
| | $(64) \rightarrow (32)$ | FC-(64, 32) |
| | $(32) \rightarrow (n)$ | FC-(32, $n$) |

Table 5: Network architectures of Graph Reasoning Network $GRN$. ($l$ is the length of node feature, $m_n$ is the number of nodes in each SCG, $m_e$ is the number of edges in each SCG, and $n$ is the number of classes of interest)

**Training details** We train $GRN$ in an end-to-end manner. Below are the details: we use Adam with $\beta_1$=0.9 and $\beta_2$=0.999, batch size 32, learning rate 0.01 for the first 25 epochs and use a decay rate of 0.8 for every 25 epochs. We train GRN for 100 epochs.

### B.6 COMPUTE RESOURCES

We use 2 RTX-2080 and 2 Telsa V100. We train knowledge distillation on RTX-2080, which costs around 2 hours for 100 epochs on RTX-2080, and we train GRN on Telsa V100, which costs 1 hour for 100 epochs.

## C MORE DETAILS OF OBJECT DATASET

OBJECT is a computer-generated 3D object image dataset using Blender Community (2018) as the rendering engine. Each image contains a 3D main object which consists of multiple basic object parts: ball, cylinder cubic, etc. Each main object is rendered using 4 independent generating attributes: object size, background color, rotation angle, sub-object material. The image size is 512 x 512. The first version of the dataset contains 13 different main objects and each one has more than 400 images. We also publicly distribute the source code, which allows one to render new data with custom main objects. The domain of attributes and resolution depends on the needed dataset size. Fig. 12 shows an example that also illustrates the workflow of new object images generation: User can select the basic objects to form the main object, we will output the corresponding three-view drawing helping people to make sure the main object configuration and parameters are correct. After the user confirms the main object configuration, we can automatically synthesize over 400 images for user defined main object with different rendering attributes mentioned above. The output datasets will contain all possible combinations of the attributes. Our primary motivation for creating the OBJECT datasets is that it allows fast testing and idea iteration, on zero-shot learning and 3D object recognition.

As there are two experiments of zero-shot learning with HNI, we upload their related datasets. There are four folders in our dataset zip file. Folders with the "main_paper" prefix mean they are the data for the main paper experiment, and folders with the "appendix" prefix mean they are the data for the appendix experiment.

You can download the dataset and its generating code from `http://anon-blind-submission`, which we plan to keep up-to-date with contributions from ourselves and the community.

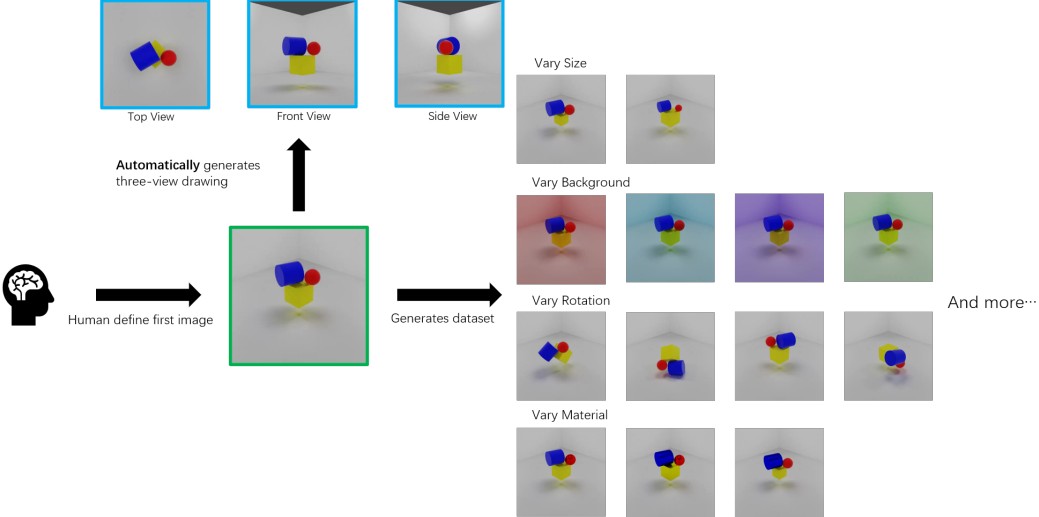

Figure 12: Overview of OBJECT dataset

## D LARGER EXPERIMENTS ON NODE AND EDGE MODIFICATION

We conducted a larger experiment with ImageNet dataset images. In addition to the original 20 classes, we randomly selected another 100 classes and constructed a 120 classes dataset. For each class, we randomly selected 300 images for training and 200 images for testing. The dataset contains

a total of 60000 images. After training, human users chose 16 classes of interest, and used the NN-to-Human path to visualize the reasoning logic (c-SCG) of NN, while the remaining classes were not modified. Then the human users analyzed the c-SCG and modified both nodes (visual concepts) and edges (Concept relationships), one human intervention per class. The performance of modified classes is shown in Table. 6 below:

Table 6: ImageNet 120 class classification performance with nodes and edges modification via HNI

| Category | american egret | black stork | spoon bill | white stork | cheetah | leopard | lion | tiger |
|---|---|---|---|---|---|---|---|---|
| Original NN Accuracy | 0.51 | 0.31 | 0.70 | 0.88 | 0.48 | 0.85 | 0.33 | 0.50 |
| Modified NN Accuracy | 0.58 | 0.40 | 0.75 | 0.92 | 0.57 | 0.89 | 0.41 | 0.54 |
| Category | ambu lance | beach wagon | cab | fire engine | jeep | limousine | recreation vehicle | school bus |
| Original NN Accuracy | 0.52 | 0.45 | 0.38 | 0.59 | 0.63 | 0.33 | 0.66 | 0.47 |
| Modified NN Accuracy | 0.57 | 0.54 | 0.44 | 0.63 | 0.70 | 0.40 | 0.75 | 0.59 |

For those 104 unmodified classes, their accuracy before and after distillation is consistent (with difference <0.03), their mean accuracy are: (42.6% before distillation and 42.9% after distillation)

As shown in Fig. 13, In our larger experiments of 120 class ImageNet experiements, for the bird "white stork" (Fig. 13(a)), we delete the edges between head and feathers because they don't have a very stable relationship, i.e. when the bird is in different poses: sit, stand, fly (Fig. 13(b)), its head and feathers may have different spatial relationships (no strong structure relationship between the two concepts). Instead, we add edges between the head and neck since it was not initially captured by the NN and they actually share reliable structural relationships. Similarly, for the "leopard"(Fig. 13(c)(d)), we delete the edges between the tail and legs and add the edges between head and back.

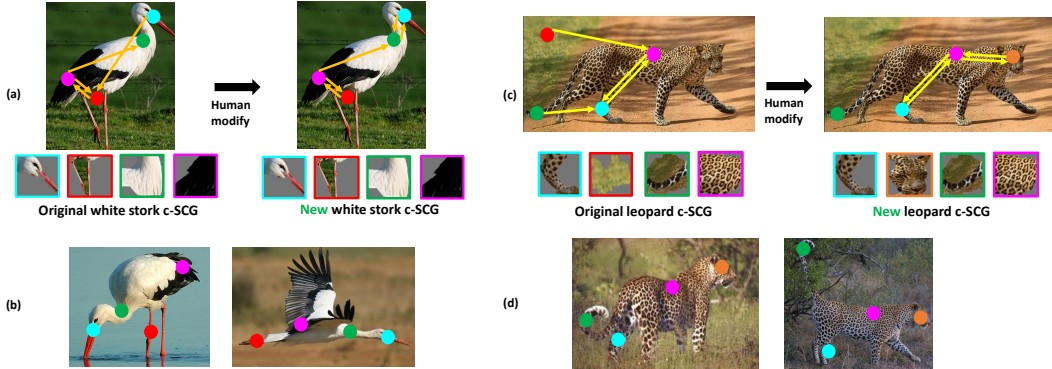

Figure 13: Examples of human modify both nodes and edges. (a) For the bird "white stork" , we delete the edges between head and feathers because they don't have a very stable relationship, i.e. (b) when the bird is in different poses: sit, stand, fly, its head and feathers may have different spatial relationships (no strong structure relationship between the two concepts). Instead, we add edges between the head and neck since it was not initially captured by the NN and they actually share reliable structural relationships. (c) and (d) Similarly, for the "leopard", we delete the edges between the tail and legs and add the edges between head and back.

# E    MORE EXPERIMENTS ON ZERO-SHOT LEARNING THROUGH HNI

We conduct two zero-shot learning experiments with HNI (Fig. 15and Fig. 14), which is similar to the Sec. 4.2 experiment of zero-shot learning in the main paper. Here, each learned object A, B, C, and D has 192 training images and 216 testing images while the new object E (we want to learn) has only 216 testing images. We use a ResNet-18 with no pretrain as classification model. Fig. 14 and Table. 7 shows the first experiments.

Different from the previous experiment, in the following experiments (Fig. 15), we have 4 different shared concepts in this experiment, and each object uses 3 of them. This setting enlarges the variance

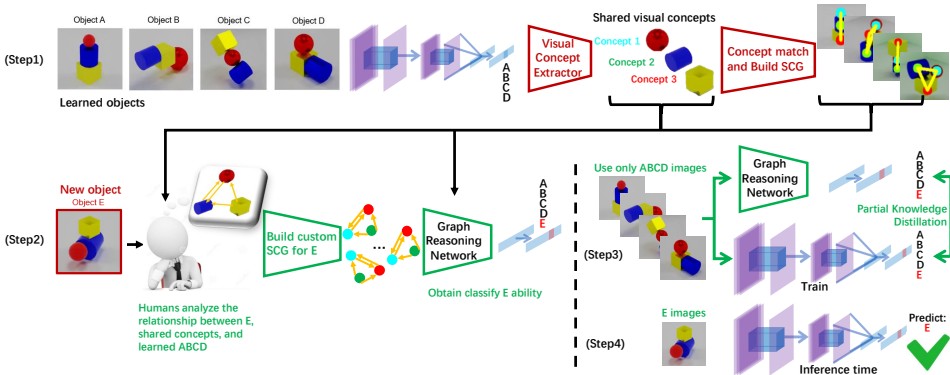

Figure 14: Zero-shot learning: Human users teach NN to learn to encode new objects with HNI. Given images of objects A, B, C, and D (object E is a new class with no images). Step 1: train a ResNet-18 with images of objects A, B, C, and D; then use VCE to discover the shared concepts; we can use concept match to obtain image-level SCGs (I-SCG) for learned objects. Step 2: humans first analyze the relationship of new object E with shared concepts and learned objects (ABCD), then build custom I-SCG for object E by recombining the nodes and edges. Then train a GRN that can classify both ABCD and E. Step 3, we use knowledge distillation to transfer the knowledge from SCG back to the original NN. Step 4: The original NN obtain the ability to classify the new object E

Table 7: Performance (Confusion matrix) of Zero-shot learning with HNI. (left) GRN with custom I-SCGs of new object E. (middle) Original NN ResNet-18 trained with images of objects A, B, C, D can not identify object E in the test set. (right) ResNet-18 learned to encode and recognize object E after knowledge distillation through proposed HNI.

| GT \ Pred | GRN A | B | C | D | E | Original ResNet-18 A | B | C | D | E | Human modified ResNet-18 A | B | C | D | E |
|---|---|---|---|---|---|---|---|---|---|---|---|---|---|---|---|
| A | 156 | 20 | 9 | 31 | 0 | 86 | 69 | 42 | 19 | 0 | **128** | 0 | 56 | 32 | 0 |
| B | 0 | 191 | 0 | 25 | 0 | 7 | 209 | 0 | 0 | 0 | 0 | **160** | 2 | 21 | 33 |
| C | 2 | 2 | 140 | 69 | 3 | 72 | 89 | 55 | 0 | 0 | 45 | 0 | **100** | 1 | 70 |
| D | 1 | 4 | 21 | 142 | 48 | 56 | 81 | 20 | 59 | 0 | 0 | 28 | 20 | **128** | 40 |
| E | 1 | 2 | 11 | 3 | 199 | 33 | 172 | 11 | 0 | 0 | 0 | 25 | 26 | 27 | **138** |

between main objects, which is similar to more general real-life settings that there may have large inter-object variance. The new object E does not need to have totally the same concepts as the training objects, such as object A, they only have two same concepts. This means that knowledge can be transferred between large variance objects which is more flexible.
Fig. 15 and Table. 8 shows the performance of zero-shot learning with our HNI.

Table 8: Performance of Zero-shot learning with HNI. (a) Confusion matrix of GRN with custom I-SCGs of new object E. (b) Confusion matrix of original ResNet-18 trained with images of object ABCD can not identify object E in the test set. (c) Confusion matrix of ResNet-18 after knowledge distillation which obtains the ability to classify new object E after human's teaching through HNI

| GT \ Pred | GNN A | B | C | D | E | Original ResNet-18 A | B | C | D | E | After distillation A | B | C | D | E |
|---|---|---|---|---|---|---|---|---|---|---|---|---|---|---|---|
| A | 130 | 83 | 0 | 0 | 3 | 161 | 51 | 0 | 4 | 0 | **160** | 56 | 0 | 0 | 0 |
| B | 0 | 216 | 0 | 0 | 0 | 0 | 216 | 0 | 0 | 0 | 67 | **149** | 0 | 0 | 0 |
| C | 0 | 0 | 165 | 43 | 8 | 40 | 0 | 105 | 71 | 0 | 13 | 71 | **115** | 0 | 17 |
| D | 0 | 0 | 13 | 203 | 0 | 0 | 6 | 0 | 210 | 0 | 0 | 0 | 24 | **192** | 0 |
| E | 0 | 14 | 1 | 0 | 201 | 42 | 11 | 0 | 163 | 0 | 0 | 55 | 48 | 26 | **87** |

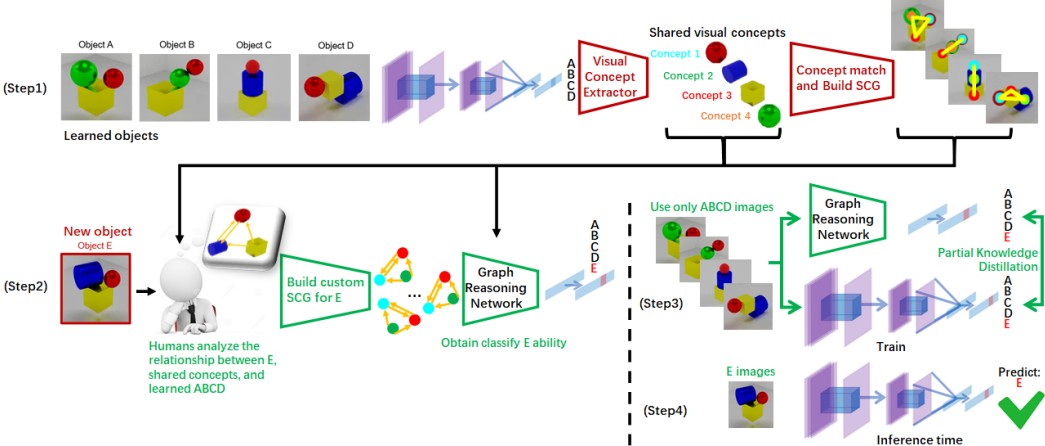

Figure 15: Configuration of general setting of Zero-shot-learning

## F HUMAN-IN-THE-LOOP USER STUDY

We conducted a user study with responses from 43 ML practitioners and students giving us the following data.

For the practitioners: (1) they are volunteers to join the human-in-the-loop User Study and it is unpaid. (2) the age range is 23 to 30 years old and there are 23 males and 20 females among all 43 participants. (3) The average time to finish this questionnaire is about 5 minutes.

Over 95% of the participants found "how to modify the Structural Concept Graph" is easy to understand. Over 93% of the participants found our method is "more helpful" than traditional ways to modify the logic of the neural network. Given these visualizations, 91% "expressed interest" in potentially using HNI to modify their models. Table. 9 shows the details of our questionnaire. As for the "explanation" referred to in question 1, we directly use Fig. 3 in the main paper as the example.

Table 9: Questionnaire details

| id | Question |
|----|----------|
| 1 | Can you understand our explanation about how to modify the Structural Concept Graph above? |
| 2 | Do you think this tool could help you modify the logic of neural network easier compared with traditional methods, such as data augmentation, modify parameters, or change the architecture of models? |
| 3 | Do you want to use this interface tool to improve the performance of your neural network? |

## G MORE DISCUSSION

### G.1 TOWARDS GENERIC INTERFACE

As this work is the first attempt towards a generic interface between human and neural networks, we primarily focus on image-related applications in our experiments. However, we want to point out that the proposed framework can be easily extended to other data modalities and tasks with minor adaptations. The input modality is only related to the concept extraction where the concepts can be extracted from other modalities such as texts and other structured data.

For regression tasks, our pipeline can easily be adapted because we do not have assumptions or constraints on the output format, the Graph reasoning network can also do regression tasks if the original NN is used to solve the regression problem. In both classification and regression tasks, humans only need to be involved with class-specific SCG (c-SCG) to modify the reasoning logic, which is agnostic to the output format. For textual input, only adaptation required is in the concept extraction, one will need to discover the concepts with different representations (e.g., important tokens).

We do have the plan to apply the proposed framework to tackle more applications such as few-shot learning, continual learning, active learning, more humanoid Neural networks, and fine-grained classification

## G.2   LIMITED DATASET SETTING

The reason we use a limited dataset is to simulate the scenario where NN fails to generalize from limited training data, hence it is easier to demonstrate how human guidance can help to improve the original NN in this case. However, we want to emphasize that the proposed framework is not limited by or making any assumption on, the training data size. First, the size of GRN depends on the classes of interest whose logic we want to modify, and it is compatible with any scale original NN with a large number of unmodified classes. As is shown in Table. 1, where we modify the logic of 6 classes of interest among 20 classes, the number of modified classes is independent of the total number of classes. We only create a Graph reasoning network for the classes of interest, and change the "local logic" where these classes may be confused with each other. On the other hand, when sufficient sampling of a large variety of classes (e.g., ImageNet) is available in training, NN is likely to be able to learn a good representation and reliable reasoning logic. In this case, NN can help humans to better understand and learn to differentiate specific classes by NN-to-Human path via the visualization of the decision logic (e.g., the most discriminative concepts and relationships). However, in real-world scenarios, if we have very limited data and a limited number of very specific classes of interest, NN may learn some hard-to-decouple bias and may fail to generalize. Human users can easily spot the bias (irrelevant concepts and incorrect relationships) and help modify the reasoning logic to improve the original NN's performance. That is the reason why, to demonstrate the efficacy of our framework, we chose to use a limited dataset to simulate the scenarios with data scarcity.

## G.3   PERFORMANCE DROP AFTER PARTIAL KNOWLEDGE DISTILLATION

GRN accuracy on the modified classes is better than modified NN after knowledge distillation. Here are the analysis and reasons:

(1) GRN is a specialist model which is only trained on modified classes of interest and only needs to consider the local logic. It cannot predict the unmodified classes. However, the original NN is a larger and general model which can classify more categories.

(2) Our proposed "Partial knowledge distillation" method (Fig. 5) tries to transfer knowledge between Graph neural network and Convolutional Neural network with different input modalities, we also have two teacher models and combine them together as soft labels. It is known that knowledge distillation is sensitive to the hyperparameter temperature ($T$). We have three different choices of $T$s in our case which makes it even harder to find the best one. Due to the goal of verifying the key idea, we do not take too much effort to conduct a hyperparameter search which may further improve the distillation performance. Moreover, we need to keep the high performance of the unmodified classes, which may also influence the performance of modified classes during knowledge distillation.

(3) Combining both GRN and original NN may have the best performance, but at an additional cost (both in parameters and run time). The reason we want to distill the knowledge from GRN back to the original NN is to try to avoid modifying the structure, size, and inference time of the original NN and instead serve as a tool to improve its performance. This way it does not affect the deployment of the original NN (e.g., the original NN may be a tiny network to be deployed on an edge platform). Our main goal is to use the proposed Human-NN Interface, to exchange knowledge between human users and original NN.

Table 10: ImageNet 20 class classification performance details (GRN accuracy) with node modification via HNI

| Category | Beach wagon | Ambu lance | School bus | Jeep | Fire engine | Recrea vehicle |
|---|---|---|---|---|---|---|
| Original NN Accuracy | 0.54 | 0.69 | 0.80 | 0.53 | 0.73 | 0.68 |
| GRN Accuracy | 0.88 | 0.86 | 0.96 | 0.93 | 0.84 | 0.80 |

Table 11: iLab-20M three class classifier performance details (GRN accuracy)

| Category | Bus | Military | Tank |
|---|---|---|---|
| Original NN Accuracy | 0.31 | 0.49 | 0.66 |
| GRN Accuracy | 0.78 | 0.91 | 0.75 |

### G.4 LIMITATIONS

(1) The zero-shot learning experiments use a synthesized dataset which is an ideal situation to show the idea. But in the real world, the shared concepts between objects may have larger variance, which may hinder the performance. (2) The knowledge distillation result is sensitive to the temperature of knowledge distillation, finding a suitable temperature is a time-consuming process. (3) The consistency of discovered concepts may not be good. However, our work highly relies on the result of discovered concepts. In order to solve this, we need to use different segmentation methods to find consistent concepts, which takes a longer time. We plan to use better methods for segmentation and match during the concept discovery and matching stage.

### G.5 BROADER IMPACT

HNI allows humans and NN to understand each other using SCG as a "language". A human can directly modify NN with human prior knowledge. It can also help us "teach" NN to learn new objects. This can have a positive impact on: (1) We can teach NN not to learn some knowledge we do not want it to learn, e.g., not learn some social bias; (2) we can help NN learn something it has never seen before. This will be useful when we lack the data. We can use our prior knowledge to build SCG and help NN to learn the knowledge of this object.
However, those who have bad intentions may be able to figure out a way to use it maliciously. In our case, one might find a way to teach NN something unsuitable. There are more and more scenarios using NN to help humans to make decisions. If we teach NN unsuitable knowledge, it may make wrong decisions, but that could be done through standard training as well.

### G.6 ASSET

Imagenet license: Researcher shall use the Database only for non-commercial research and educational purposes
We use a subset of the ILSVRC2012 dataset (ImageNet) and iLab-20M. The objects in ImageNet are vehicles and animals that can be found on the Internet. The object in iLab-20M is toy vehicles, under Creative Commons CC-BY license. They do not contain any personally identifiable information offensive content.

