# OpenReview forum: "Towards Generic Interface for Human-Neural Network Knowledge Exchange"
_ICLR.cc/2022/Conference — ICLR 2022 Submitted_

### Official Review · Reviewer_GNsq · 2021-10-25

**Correctness:** 4
**Technical Novelty And Significance:** 3
**Empirical Novelty And Significance:** 2
**Recommendation:** 5
**Confidence:** 3

**Main Review:**

Strengths:
-  The idea is an interesting one, the paper does a good job of building on prior work, and the methodology seems sound.
-  The results show that this method is useful in fairly low-data regimes.

Weaknesses:
-  Overall, the experiments don't do a great job of higlighting how the method could be useful:
	-  While low data regimes may be interesting, all of the ones studied in this paper are "artificial" in some sense:  4.1.1 -> sub-samples ImageNet, 4.1.2 -> sub-samples (in a deliberately biased way) iLAB-20M, 4.2:  uses synthetic data.
	-  The paper focuses on human interaction/intervention, but the interventions in 4.1.2 (deleting all edges) and 4.2 (automatically generate new I-SCGs) are both algorithmic and seemingly could be done (or at least tried) based soley on domain knowledge without seeing the C-SCGs/explanations.

**Summary Of The Paper:**

The paper proposes a technique for allowing humans to interact with NNs.  It starts by approximating a NN by extracting concepts from an image, using those concepts to form a graph, and then putting that graph through a GNN.  Then, it allows users to edit these "concept based graphs" and uses those edited graphs to retrain the GNN.  Experimentally, this is shown to improve performance in three different experiments.


**Summary Of The Review:**


Overall, the reviewer think that this is a ok, but not great, paper.  With a more realistic experiment (ie, with a naturally occuring dataset and interventions from users based on the explanations), the paper would be much stronger.

---

> ### Author Response · Authors · 2021-11-11
> **Response to Reviewer GNsq**
>
> Dear Reviewer,
>
> Thank you for reviewing our paper. While we are happy that you consider our idea is “interesting idea” and “does a good job”, our method is a “sound methodology” and point out our experiments “results show that this method is useful", you also mention a number of potential concerns/critical comments to which we respond below:
>
> (1) ***"Artificial" of the dataset***: Thanks for your suggestion! The reason we use a subset class (not all classes) is to simulate the scenario where NN fails to achieve perfect accuracy, hence it is easier to demonstrate how human guidance can help to improve the original NN in this case. Since most classes are already correctly classified by the base NN, here we focus on the lower-performing classes (vehicles may confuse each other), so that we can demonstrate how to fix them. Although we want to emphasize that the proposed framework is not limited by or making any assumption on, the training data size, in general, this is the way our method should be used: we only need to examine and correct the classes that have problems.
>
> (2) ***Human Interaction***: Thanks for your suggestion! Yes, our focus is on the interaction and we want to provide not only a useful but also an **efficient** user interaction experience: we want human users only do the most irreplaceable work: understanding the logic and modifying the logic through highly abstract and understandable language (structural visual concept). Once the logic modification is done, the rest of the knowledge exchange should be as automatic as possible to improve efficiency.
>
> For edge modification in Sec.4.1.2, the most important and irreplaceable interaction with human users involved is analyzing the reasoning logic. It is intuitive for the human user to identify that concepts (nodes) are correct (blue node colors), but relations (edges) are incorrect  (red edge colors) which led to misclassification. Based on these observations,  the human user removes all edges which fixes the errors.
> Similarly, In section 4.2, the reason why we can automatically generate new I-SCGs for novel class E is that human users analyze the relationship between the novel class E with the learned class (A,B,C, D) and find the shared visual concept and similar structural relationship. That is the irreplaceable job of human user deciding which learned visual concepts and concept relationships should be used to generate new I-SCG. While it may also be correct that the interventions can be done solely on domain knowledge for some cases and scenarios, we argue that the C-SCGs/explanations are still crucial because of the following reasons:
> 1) the C-SCGs are the common language understandable by both NN and human user and the communication between human user and NN based on this language is the most efficient and straightforward;
> 2) with C-SCGs human users can focus on a small number of classes which are low-performing and requires interventions.
> 3) they provide human users a starting point (instead of from scratch) to complete and correct the representations for classes of interest.

---

> > ### Author Response · Authors · 2021-11-11
> > **Further Explanations**
> >
> > We provide further explanation about your suggestion "*interventions from users based on the explanations, the paper would be much stronger.*"
> >
> > ***Interventions based on explanations***: Thanks for your suggestion! Yes, Interventions based on explanation is exactly what we want to demonstrate in our evaluation: the NN-to-Human path can provide not only global-wise explanations, class-specific Structural Concept Graph (c-SCG), but also instance-wise explanations (Sec.4.1.2 row 3~10 and Appendix Sec.A.1). Those explanations provide guidance and support for human interventions in the Human-to-NN path.
> >
> > Specifically, c-SCGs explain the reasoning logic of the original NN by highlighting the important visual concept and relationship between concepts, human users can then analyze the explained reasoning logic and make interventions based on their knowledge accordingly. At the same time, Instance-wise explanations can explain more detailed reasoning logic for any input image by answering the question: “why class A but not class B”? With the help of concrete and detailed example explanations provided by instance-wise explanations, human users can clearly understand the explanation and apply interventions more easily. For instance, as we described in the experiments of Sec.4.1.2, after knowing “military” images are easily misclassified as “tank”, only based on c-SCG of military and tank may not be obvious to the human user which part of the representation requires intervention (as on the surface it makes some sense and all important concepts on c-SCG are from the foreground). When we use instance-wise explanation to explain the decision details for some misclassified images (e.g. Fig.6 (b)), from the color of nodes and edges ( blue for positive contribution and red for negative contribution) of the decision, we can immediately understand that relations (edges) are incorrect  (edges are labeled in red) which have led to misclassification. Based on these observations,  human users can then easily apply effective interventions:  in this case removing all edges, which fixes the errors.
> >
> > Overall, we believe that the explanations make intervention much easier and effective, and encourage more knowledge exchange and support between the NN-to-Human path and the Human-to-NN path.

---

> ### Comment · Reviewer_GNsq · 2021-11-15
> **Reviewer Response to Author Comments**
>
> "We provide further explanation about your suggestion 'interventions from users based on the explanations, the paper would be much stronger.'"
>
> This quote is not representative of what the reviewer said which is "With a more realistic experiment (ie, with a naturally occuring dataset and interventions from users based on the explanations), the paper would be much stronger."
>
>
> "The reason we use a subset class (not all classes) is to simulate the scenario where NN fails to achieve perfect accuracy, hence it is easier to demonstrate how human guidance can help to improve the original NN in this case."
>
> This is consistent with the reviewer's point:  using an "artificial" setup makes it easier for a method to succeed.  Because methods that work on easier problems may not work on harder problems, it's entirely possible that the proposed method does not work on more realistic/important settings.
>
>
> "While it may also be correct that the interventions can be done solely on domain knowledge for some cases and scenarios, we argue that the C-SCGs/explanations are still crucial because of the following reasons:"
>
> Similar to (and compounded with) the above comment, the problem is that by evaluating on settings where domain knowledge (without explainations) is sufficient, it is unclear whether or not the method will still work on more realisitic/important settings where domain knowledge alone is not sufficient. It's not clear to the reviewer that the interventions in 4.1.1 (removing all edges, which seems to be a reasonable sanity check for any graph method) and 4.2 (algorithmic generation, which seems to be possible based on knowledge of this synthetic dataset) actually require anything beyond domain knowledge.

---

> > ### Author Response · Authors · 2021-11-22
> > **Response 2 to Reviewer GNsq**
> >
> > Thanks for your feedback!
> >
> > 1 **For the "artificial" setup**:
> >
> > (1) We conducted larger and more natural experiments (please refer to Top comments "Conducted more experiments and Updated paper" for more details), where we randomly choose 100 classes from ImageNet on top of the previous 20 classes in the main paper to form a 120 class dataset (60000 images total). The results demonstrate the effectiveness of the proposed framework.
> >
> > (2) Our work focuses on providing an interface between human users and NN, which can efficiently communicate with each other through the novel human and NN friendly language “Structural Concept Graph”. The experiments of human users modifying the logic of NN improving its performance do not aim to show that human logic is always better than the learned logic by NN and can always lead to better performance of NN, especially when NN has sufficient data. Indeed, the original NN is already able to classify some classes very well. Here we focus on how humans can understand the failures of other classes, and then fix them.  Thus, we wanted to show that as an interface for knowledge exchange, our proposed framework is bidirectional and always follows the basic rule that: the party with better understanding can teach and help the other party with more confusion. The application and effectiveness of our interface are different in different scenarios:
> >
> > 1) When there is enough data to train NN and NN has high accuracy (e.g., ImageNet)
> > In that case, the learned logic of NN is the party with better knowledge, as we claimed in the previous response, our method (as an interface) can show the learned reasonable logic of NN to humans which can help humans better understand some complex classes: For instance, when we have sufficient data for those obscure categories that are new or challenging for human users. Instead of human users modifying the logic of NN, NN can show their learned logic to human users and “improve” the understanding of humans on these classes of interest. Thus, the neural network first analyzes thousands of images to extract the most relevant primitive visual concepts, and can then teach them to human users through our proposed concept graph, without requiring human users to go through all the (thousands of) images and try to generalize themselves.
> >
> >  2) When there are not enough data to train NN and NN has low accuracy (e.g., experiments in Sec.4.1.1 and Sec.4.1.2)
> > In that case, NN may learn some irrelevant/incorrect logic from a small or biased training set, which can be hard to generalize well on the test set. In that case, human users may have a more robust understanding especially for classes that are often seen in everyday life. So humans can easily identify the unreasonable reasoning logic of NN and modify it. In this case, our human-to-NN path can transfer the modified logic back to NN and improve its performance. For examples, please see figures 2, 3, 13.
> >
> > 2 **For the domain knowledge concern**:
> >
> > In our new larger experiments of 120 class ImageNet experiments (Appendix Section D in page 17, 18)，for the bird “white stork” (Fig. 13 (a) in page 17) , we delete the edges between head and feathers because they don’t have a very stable relationship, i.e. (Fig. 13 (b)) when the bird is in different poses: sit, stand, fly, its head and feathers may have different spatial relationships (no strong structure relationship between the two concepts). Instead, we add edges between the head and neck since it was not initially captured by the NN and they actually share reliable structural relationships. Similarly, for the “leopard”, (Fig. 13 (c) (d)) we delete the edges between the tail and legs and add the edges between head and back. We agree that from the users’ perspective, this intervention can be done solely relying on their domain knowledge. However, we argue that without the interface we proposed in this work, it would be very challenging and cumbersome for the human users to understand the reasoning logic of the NN and identify the problem in the first place, and even more cumbersome and inefficient for them to transfer such domain knowledge back to the NN.
> >
> > In Zero-shot learning experiments： Human-in-the-loop is crucial and it takes way more than just the domain knowledge of the users (e.g., how to convert domain knowledge into constraints, supervision, regularizations, or data in training the NN). In fact, what we are trying to showcase here is that we proposed an efficient and intuitive mechanism and interface for human users to easily apply their domain knowledge in this task. The interface we proposed makes no assumption on attribute labels, does not modify the original NN’s architecture, and does not rely on manipulating training data. Instead, it relies on the “common language” based on GCN to communicate and exchange knowledge between human users and NN enabling human users to easily teach the NN to encode novel objects with their domain knowledge, effortlessly.

---

### Official Review · Reviewer_KrBb · 2021-11-02

**Correctness:** 2
**Technical Novelty And Significance:** 2
**Empirical Novelty And Significance:** 3
**Recommendation:** 5
**Confidence:** 4

**Main Review:**

The paper proposes a sensible approach for the relevant problem of interactive learning for image classification. While there are several interactive methods for CAM-based explanations, it is interesting to explore this direction for structured explanations.

The related work section covers relevant papers for human-in-the-loop learning, but is quite sparse. There are numerous other works with similar goal as the cited paper [1], also for other data modalities such as text. In addition, several other works exploit scene graphs user interaction (e.g. [3]) for different, but related goals (here image generation), which might be relevant for this section.

Given that the paper is titled “generic interface”, it would be required to cover more modalities. This might be an argument for the whole introduction / motivation of the paper, as only images are used to explain and evaluate the method.

The rest of the related work section mostly covers “components” that the approach is built on (i.e. image explainers, GNNs, Knowledge distillation) with the exception of zero-shot learning, which could be remedied for overall readability. A possibility would be a designated background section. The problem continues in the main part of the paper, where the reused explanation method [2] is cited but not clearly distinguished from the contribution (“NN-TO-HUMAN”).

The evaluation results are interesting and promising. However, a lot of useful information are buried in the end of the suppl. material. This is cumbersome, as these are basic information about dataset origins or user study setup. More specifically, it would be very helpful to add a short explanation for the origin of the object dataset in evaluation setup and clearly define the user study before presenting the results. To this end, did the mentioned user study only cover the qualitative questions about the experience with the system or also the actual manipulation of structured explanations for the quantitative results? It is not clear to me from the sentence.

Similarly, it is vital to motivate why the evaluation suffices for the paper claims, which is also not part of the main paper. The suppl. Material contains limitations of the current approach and evaluation, which cover parts of this.

For the zero-shot learning experiment, it would be insightful to add a representative zero-shot baseline to better assess the added value of human intervention. While it is good to show that the method can generalize in the block example, there are established zero-shot benchmarks which could be used at this point.

[1] Li, K., Wu, Z., Peng, K.C., Ernst, J. and Fu, Y., 2018. Tell me where to look: Guided attention inference network. In Proceedings of the IEEE Conference on Computer Vision and Pattern Recognition (pp. 9215-9223).
[2] Ge, Y., Xiao, Y., Xu, Z., Zheng, M., Karanam, S., Chen, T., Itti, L. and Wu, Z., 2021. A Peek Into the Reasoning of Neural Networks: Interpreting with Structural Visual Concepts. In Proceedings of the IEEE/CVF Conference on Computer Vision and Pattern Recognition (pp. 2195-2204).
[3] Mittal, G., Agrawal, S., Agarwal, A., Mehta, S. and Marwah, T., 2019. Interactive image generation using scene graphs. arXiv preprint arXiv:1905.03743.

**Summary Of The Paper:**

The paper proposes an interactive learning algorithm for image processing tasks based on structural explanations. The paper therefore reuses recent advances in creating structured image explanations for image classification and uses knowledge distillation to update the model after end-users manipulated them.

The approach is evaluated for robustness / predictive improvement and zero-shot learning on known datasets (ILSVRC2012 and iLab-20M, object dataset). The baselines are the non-interactive system and a system with more training data per class - each with the same architecture. The results show that the proposed approach generates better results after human corrections and can be used for novel classes.

**Summary Of The Review:**

The paper tackles a relevant problem for the conference. While the empirical evaluation is rather insightful for image processing, the paper is motivated for any data modality, but the claims are not backed up by evidence. The paper could also improve the readability / clarity of presentation.

---

> ### Author Response · Authors · 2021-11-11
> **Response to Reviewer KrBb**
>
> Dear Reviewer,
>
> Thank you for reviewing our paper. While we are happy that you consider our idea is an “interesting direction to explore”, our method is a “sensible approach” and experiments “evaluation results are interesting and promising", you also mention a number of potential concerns/critical comments to which we respond below:
>
> (1) ***Related works***: Thanks for your suggestion, we will cite these papers and discuss them in the related works.
>
> (2)***Towards generic***: Towards the generic interface, we discussed in Appendix F.1. As this work is the first attempt towards a generic interface between human and neural networks, we primarily focus on image-related applications in our experiments. However, we want to point out that the proposed framework can be easily extended to other data modalities and tasks with minor adaptations. The input modality is only related to the concept extraction where the concepts can be extracted from other modalities such as texts and other structured data.
> Nevertheless, we agree that we could delete the word “generic”  throughout as indeed all our results so far are in image classification.
>
> (3) ***Background***: Thanks for your suggestion, yes, the NN-to-Human path is built based on the cited paper [2], we take some room to explain the details because we want clearly explain this part as it is important for the readers to further understand the novel Human-to-NN path with human modification, Graph reasoning Network, partial knowledge distillation, and further applications.
>
> (4) ***OBJECT dataset***: Due to the room limitation, we put the details about our OBJECT dataset in Appendix Sec C. It contains the motivation, source code description, dataset basic information (size, quality, content), creation workflow, and so on. We will also opensource all the code and datasets upon acceptance. Please let us know if you have any other comments. Thanks!
>
> (5) ***User study***: The goal of the user study in the Appendix. Sec. E is to get a preliminary understanding of whether the interaction design of our framework is acceptable to users. The current version of user study does not contain actual manipulation of structured explanations. We agree that a more extensive, detailed user study is required in order to better understand the usability of our framework in real-world applications. A  larger-scale user study with more technical details and a clearer interface on Amazon Mechanical Turk is currently underway.
>
> (6) ***Motivate why evaluation suffices for claims***: To demonstrate that HNI can be used as a generic interface for knowledge exchange between human users and NN. We want to show humans can transfer their knowledge to NN through our HNI which supports our above claim. We use three main experiments: the first two are human users transferring their knowledge to NN in order to correct their original logic and further improve the performance of the original NN. (Sec. 4.1.1 by node modification and Sec. 4.1.2 by edge modification). The third experiment is to demonstrate through HNI, a human user can transfer their understanding of new classes to help NN achieve zero-shot learning (Sec. 4.2), where humans can guide NN in learning to encode new objects from unseen categories. The success of the aforementioned three tasks not only shows the knowledge transfer from human to NN but also NN to human because NN to human is the foundation: human users should understand NN’s reasoning logic first before transferring knowledge to NN. The experiments also show that with our HNI, which is a novel knowledge exchange pipeline taking structural concept graph as “communication language”,  we can design new solutions for many tasks such as continual learning,  few-shot learning. We argue that this is a small but important step forward towards a generic interface for interaction between humans and NN.
>
> (7) ***zero-shot learning***: Thanks for your suggestion! we looked at many zero-shot learning methods. We tried but could not find a suitable baseline. Indeed, typical zero-shot baselines define new classes from attribute vectors (e.g., an object with horse features and stripes is defined as a zebra). In our example in Fig. 7, object E has all the same features as objects A, B, C, and D: all have a red sphere, blue cylinder, and yellow cube. The difference is in the configuration, captured by our structural graph. While we could not find an existing dataset that would allow for defining new objects solely based on structure, we believe that this is a great idea, and creating such a dataset could advance zero-shot research significantly. In the meantime, we consider it an advantage of our method to be able to define new objects by either parts (concepts), relationships (structure graph), or both.
> We discussed the limitations of our method in Appendix F.4: in the real world, the shared concepts between objects may have a larger variance, which may hinder the performance.

---

### Official Review · Reviewer_9inm · 2021-11-03

**Correctness:** 3
**Technical Novelty And Significance:** 3
**Empirical Novelty And Significance:** 2
**Recommendation:** 6
**Confidence:** 3

**Main Review:**

Whenever we train or fine-tune a neural network if it's going to be used in an application that is safety-critical the resulting network must go through a thorough investigation to make sure that it performs as expected in all classes and this would be a time-consuming process. Whenever we train a new student network, there are many things that could go wrong for example a bad learning rate, bad choice of hyper-parameters. Investigating the accuracy and behavior of the neural network for unexpected behaviors, vulnerability to adversarial attacks, etc. is time-consuming. Every time we make a small change in the decision-making process of one of the classes using HNI, such a verification process should be repeated. This sounds a little impractical.

I think the application of the proposed method can be better justified for training neural networks when there are not enough training samples or in zero-shot learning. The paper has provided experimental results related to such scenarios which is good.
Modifying nodes (adding or removing concepts) seems to be much more intuitive than modifying edges. I found that modifying edges is not very intuitive. For example, in section 4.1.2 it is not clear how a human should know that's removing all edges will result in removing the effect of pose.


**Summary Of The Paper:**

The paper provides a way for explaining the reasoning of a neural network to humans in the form of a class-specific structural concept graph (c-SCG). The c-SCG can be modified by humans. The modified c-SCG can be incorporated in training a new student model. Experiments show that the new model performs better on classes that their corresponding c-SCG have been modified.

**Summary Of The Review:**

The neural network-to-human path heavily overlaps with (Ge et. al 201). I think the novelty of the work is mostly in the human-to-NN path.

---

> ### Author Response · Authors · 2021-11-11
> **Response to Reviewer 9inm**
>
> Dear Reviewer,
>
> Thank you very much for your positive review and your valuable feedback! It is very encouraging that you rate our work as “novel”, “intuitive” and “good experimental results”.
> Please find our responses to your points below:
>
> (1) ***Verification process***: Thanks for your suggestion!  While we agree the verification process is important, we consider it orthogonal to our method because it is a common requirement for any method that creates/modifies/retrains/fine-tunes a new neural network. We agree that more work should be done to make verification more efficient, for any method, and that this is a very interesting direction for future research. In our method, one advantage is that after understanding the reasoning logic of NN through the NN-to-human path, a human user can modify many classes at the same time (e.g., in Table 1 we modify 6 vehicles), and after the whole knowledge exchange through the human-to-NN path, we only need one verification for the obtained modified NN. In other words, the number of modified classes is not related to the number of verifications required.
>
> (2) ***Edge modification***: Edges in the Structural concept graph (SCG) represent structural relationships between parts. Our method highlights those (e.g., in Fig. 3 (a) the equipment (pink) should be above the bumper (blue)). When the network makes a mistake, our approach shows the contribution of every node and edge to the final result through instance-wise explanation (Sec.4.1.2 row 3~10 and Appendix Sec.A.1). Thus, humans can inspect those contributions and judge whether these edges (relationship between concepts) are reasonable. For example, in Fig. 3, the tiger’s tail was linked to the hind leg with high importance, but a human user can immediately recognize that this relationship (derived from the original network) does not always hold and decide to remove it (or at least reduce its importance). Thus, we argue that modifying edges is also quite intuitive. We will add more discussions and clarifications in the revised version. For the example in 4.1.2, the basic idea is that concepts (nodes) were correct (blue node colors), but relations (edges) were wrong (red edge colors) which led to misclassification. Removing all edges in this case corrected the errors.

---

> > ### Author Response · Authors · 2021-11-22
> > **Response 2 to Reviewer 9inm**
> >
> > Dear Reviewer,
> >
> > We added two new larger experiments in the updated paper (please refer to top comments “Conducted more experiments and Updated paper” for more details).
> >
> > In our new larger experiments of 120 class ImageNet experiments (Appendix Section D on pages 17, 18), we provide more edge modification examples: for the bird “white stork” (Fig. 13 (a) on page 17), we delete the edges between head and feathers because they don’t have a very stable relationship, i.e. (Fig. 13 (b)) when the bird is in different poses: sit, stand, fly, its head and feathers may have different spatial relationships (no strong structure relationship between the two concepts). Instead, we add edges between the head and neck since it was not initially captured by the NN and they actually share reliable structural relationships. Similarly, for the “leopard”, (Fig. 13 (c) (d)) we delete the edges between the tail and legs and add the edges between head and back.

---

### Official Review · Reviewer_yaEq · 2021-11-14

**Correctness:** 3
**Technical Novelty And Significance:** 3
**Empirical Novelty And Significance:** 3
**Recommendation:** 6
**Confidence:** 4

**Main Review:**

Strength

	1. A new framework to connect the explainable neural net paradigm to a paradigm that can make use of the explanation to allow human intervene the neural net models. A success in this direction will open up a new era of data-driven machine learning in a long run.

Weakness

	1.  The fidelity of the SCG lacks theoretical guarantee or at least lack systematic after-math accounting means. This is inherited from VRX (Ge et al. 2021) and the authors did not address it in this paper except for human review.
	2. Similarly, there is a need for high fidelity accounting  on human-to-nn path regarding whether the distilled NNs truly reflect the  human's modification on the reasoning of NN explanation. Indeed, in page 8, deleting all edges are used to train GRN with the new c-SCG.
	3. The experiments are conducted at the scale of 200 images and 20 classes. It is not clear whether the proposal will still observe the same performance gains for a large scale datasets and a large number of classes.

Details:

	1. Have you tried a large dataset with a larger number of classes? Please do ablation study at scale to understand the scaling properties of your approach.
	2. Page 8, how "deleting" all edges compares to the original modified c-SCG? Why wasn't it working in the first place? Does it indicate that the c-SCG graph itself did not convey the right format or the right logic in full which can recover the NN behavior in the original tasks? This is worrisome as it might meant that the original VRX (Ge et al. 2021) might have fidelity flaw in its methodology in the first place.
	3. For zero-shot experiments, 4 seen classes and 1 unsee classes over a 192 training images per seen class is too small a dataset to reveal the possible success or weakness of the approach.


**Summary Of The Paper:**

This paper extended VRX (Ge et al. 2021) with an interface to allow human modify the class-specific structural concept graphs (c-SCG) as well as a procedure to distill the human changes in the c-SCGs back to original task's neural networks. This is an interesting piloting idea. However, the scale of the experiments is not convincing enough to demonstrate the generalizability of the performance to a large number of concepts, classes, datasets and complex scenarios.

**Summary Of The Review:**

This might be a game-changing direction for the future however the paper as it is now is lacking solid ground and mature enough progress in this direction.

---

> ### Author Response · Authors · 2021-11-15
> **Response to Reviewer yaEq**
>
> Dear Reviewer,
>
> Thank you very much for your positive review and your valuable feedback! It is very encouraging that you rate our work as “an interesting piloting idea”, “game-changing direction for the future” and “may open up a new era of data-driven machine learning in a long run.”
>
> Please find our responses to your points below (ordered by topic):
>
> (1) **Page 8 experiments** (Details (2)): We apologize for the confusion. In fact, there is only one modification done by human users in that experiment. The edge-editing experiment (Sec. 4.1.1, conducted on ImageNet dataset) was independent of the concept-editing experiment (Sec. 4.1.2 conducted on iLab-20M dataset). The original c-SCG (before deleting all edges) is able to convey the logic of the original NN and the errors identified by human users (incorrect relationships / edges between concepts) are likely due to the bias in the training set, which is why the performance of the original NN is poor in the first place (Fig.6 (c) row 1).
>
> (2) **Fidelity accounting on human-to-nn path** (Weakness (2)):  Thanks for your suggestion! Here, we show the results of Graph Reasoning Network after independent training with human-modified c-SCG. After partial knowledge distillation, we get the modified NN.
>
> |   Category        | Bus | Military | Tank |
> |:----| :----:| :----: | :----: |
> |Original NN Accuracy | 0.31 | 0.49 | 0.66 |
> |GRN Accuracy            | 0.78 | 0.91 | 0.75 |
> |Modified NN Accuracy| 0.73 | 0.85 | 0.81 |
>
> (3) **Larger scale experiments** (Weakness (3) and Details (1)): Thanks for your suggestion! We discussed the motivation of experiment setting in Appendix Sec. F.2, here are the details:
> The reason we use a limited dataset is to simulate the scenario where NN fails to generalize from limited training data, hence it is easier to demonstrate how human guidance can help to improve the original NN in this case. However, we want to emphasize that the proposed framework is not limited by or making any assumption on, the training data size. First, the size of GRN depends on the classes of interest whose logic we want to modify, and it is compatible with any scale original NN with a large number of unmodified classes. As is shown in Table. 1, where we modify the logic of 6 classes of interest among 20 classes, the number of modified classes is independent of the total number of classes. We only create a Graph reasoning network for the classes of interest and change the “local logic” where these classes may be confused with each other. On the other hand, when sufficient sampling of a large variety of classes (e.g., ImageNet) is available in training, NN is likely to be able to learn a good representation and reliable reasoning logic. In this case, NN can help humans to better understand and learn to differentiate specific classes by NN-to-Human path via the visualization of the decision logic (e.g., the most discriminative concepts and relationships). However, in real-world scenarios, if we have very limited data and a limited number of very specific classes of interest, NN may learn some hard-to-decouple bias and may fail to generalize. Human users can easily spot the bias (irrelevant concepts and incorrect relationships) and help modify the reasoning logic to improve the original NN’s performance. That is the reason why, to demonstrate the efficacy of our framework, we chose to use a limited dataset to simulate the scenarios with data scarcity.
>
> To further address the concerns, we are conducting a larger scale experiment with more classes and we will report the results once we finish or in the final version. Thanks!
>
>
> (4) **Zero-shot experiments** (Details (3)): Thanks for your suggestion! We are conducting a larger scale experiment and we will report the results once we finish or in the final version. Thanks!

---

> > ### Author Response · Authors · 2021-11-15
> > **Further Response**
> >
> > **Fidelity of SCG** (weakness (1)) : Thanks for your suggestion! Yes, we discussed the fidelity of using SCG to effectively reveal the reasoning logic at the end of Sec. 3.1 and Appendix Sec. A.2.2. Here are the details: While we agree that our current pipeline does not provide a guarantee that SCG will be 100% faithful to the original NN’s reasoning logic, we would like to point out a few considerations in our design:
> >
> > (1) The faithfulness of our SCG to the original NN depends on several factors, including the concept selection, the number of concepts used to construct the SCG, the sampling of data used in distillation from the original NN to SCG. Since the focus of this paper is more of a feasibility study, rather than a ready-to-deploy system, each component inevitably has errors to certain extent.
> >
> > (2) We use top-down attention (e.g. Grad-CAM [1]) to identify regions in the input image which contributed most to the prediction of each class-of-interest, which can limit the region in which we conduct concepts discovery. By doing this, we can filter irrelevant concepts and focus on those considered important in the original NN’s reasoning/inference.
> >
> > (3) In concept selection, we use concept activation vectors (TCAV) [2] to compute the importance score of each concept w.r.t. the prediction of a specific class-of-interest. With sufficient sampling of data, this gradient-based method can faithfully reflect the overall correlation between each visual concept and each class-of-interest in the original NN.
> >
> > (4) Knowledge distillation is widely used in transferring knowledge (encapsulating the reasoning logics) between neural networks. In other words, it is transferring the input-output mapping from one neural network to another. In our framework, we adapted knowledge distillation to imitate such mapping (instead of mapping from the input image to output logits, we learn the mapping between the visual concepts and the output logits). We also add perturbation during distillation by randomly masking out one of the detected visual concepts in the input image while making sure GRN and original NN produce the same outputs. Given enough training data, GRN is expected to faithfully mimic the original NN and reveal the reasoning logics.
> >
> > (5) Experimental validation. The faithfulness of using Structural Concept Graph (SCG) to explain the original NN is one key contribution of Visual Reasoning Explanation (VRX) ref [3], and in the paper, the authors conducted several experiments to prove the fidelity of SCG explanation, such as logical consistency between the explanation using SCG and the original NN. To briefly recap:
> >
> > 1) In Sec.4.2 of VRX paper [3], they qualitatively and quantitatively verified the logical consistency between explanation using SCG and original NN by showing that VRX can correctly locate the reason that causes the original model’s wrong prediction and successfully correct the errors with the guidance of the explanation (Table 1 of VRX paper [3] shows 114/119 errors were successfully corrected). The logical consistency verification experiment shows that the explanation with SCG is faithful to the original NN.
> > 2) Explanation sensitivity of visual and structural perturbations. The authors also conducted two experiments with the control variate method to verify the explanations of VRX are faithfully and sensitive to visual perturbation and structural perturbation respectively (Sec.4.3 and Figure 7 of VRX paper [3] ). Based on the above-mentioned designs and experimental validations on the fidelity of SCG explanations, we argue that the explanation with SCG can be faithful to the original NN, in ideal scenarios (assuming every component’s error is minimized). As part of future work, we plan to investigate the performance influence of the non-perfect fidelity of SCG. However with our current pipeline, the users can rely on the knowledge distillation loss, which can serve as a good indication of the faithfulness of the SCG to the original NN, to gain a better understanding of whether the SCG needs further improvement (e.g. with more data used in knowledge distillation, more visual concepts should be selected, etc.)
> >
> > Reference
> >
> > [1] Selvaraju, Ramprasaath R., et al. "Grad-cam: Visual explanations from deep networks via gradient-based localization." Proceedings of the IEEE international conference on computer vision. 2017.
> > [2] Kim, Been, et al. "Interpretability beyond feature attribution: Quantitative testing with concept activation vectors (tcav)." International conference on machine learning. PMLR, 2018.
> > [3] Ge, Yunhao, et al. "A Peek Into the Reasoning of Neural Networks: Interpreting with Structural Visual Concepts." Proceedings of the IEEE/CVF Conference on Computer Vision and Pattern Recognition. 2021.

---

> > > ### Author Response · Authors · 2021-11-22
> > > **Additional experiments updated**
> > >
> > > Dear Reviewer,
> > >
> > > Thanks for your constructive comments!  We have added additional larger-scale experiments in the paper. In particular,  we ran a Larger experiments of node and edge modification and a Larger experiments of Zero-Shot learning: Here are the details:
> > >
> > > (1) **Larger experiments of node and edge modification**: (updated paper Appendix Section D) We conducted a larger experiment with ImageNet dataset images. In addition to the original 20 classes, we randomly selected another 100 classes and constructed a 120 classes dataset. For each class, we randomly selected 300 images for training and 200 images for testing. The dataset contains a total of 60000 images. After training, human users chose 16 classes of interest, and used the NN-to-Human path to visualize the reasoning logic (c-SCG) of NN, while the remaining classes were not modified. Then the human users analyzed the c-SCG and modified both nodes (visual concepts) and edges (Concept relationships), one human intervention per class.  The performance of modified classes is shown in the table below:
> > >
> > >
> > > |   Category        | american_egret| black_stork| spoonbill|white_stork|cheetah|leopard|lion|tiger|
> > > |:----| :----:| :----: | :----: | :----:| :----: | :----: | :----:| :----: |
> > > |Original NN Acc. | 0.51 | 0.31 | 0.70 | 0.88 | 0.48 | 0.85 | 0.33 | 0.50 |
> > > |Modified NN Acc.| 0.58 | 0.40 | 0.75 | 0.92 | 0.57 | 0.89 | 0.41 | 0.54 |
> > >
> > > |   Category        | ambulance| beach_wagon| cab|fire_engine|jeep|limousine|recreation_vehicle|school_bus|
> > > |:----| :----:| :----: | :----: | :----:| :----: | :----: | :----:| :----: |
> > > |Original NN Acc. | 0.52 | 0.45 | 0.38 | 0.59 | 0.63 | 0.33 | 0.66 | 0.47 |
> > > |Modified NN Acc.| 0.57 | 0.54 | 0.44 | 0.63 | 0.70 | 0.40 | 0.75 | 0.59 |
> > >
> > > For those 104 unmodified classes, their accuracy before and after distillation is consistent (with difference <0.03), their mean accuracy are: (42.6% before distillation and 42.9% after distillation)
> > >
> > > (2) **Larger experiments of Zero-Shot learning** (updated paper Section 4.2) : We conducted a larger experiment with OBJECT dataset: We chose 11 objects from OBJECT dataset. 8 of them are seen classes (class A to H) and each class contains 300 training images and 216 testing images. 3 of them are unseen novel classes (class I, J, K) and each unseen class contains only 216 test images. The dataset contains a total of 4776 images.
> > >
> > > The table below shows the results:
> > >
> > >
> > > |   Category        | A| B| C|D|E|F|G|H|I|J|K|
> > > |:----| :----:| :----: | :----: | :----:| :----: | :----: | :----:| :----: |  :----: | :----:| :----: |
> > > |Original NN Accuracy | 0.8 | 0.79 | 0.9 | 0.74 | 0.87 | 0.78 | 0.9 | 0.9 |0 | 0 | 0 |
> > > |Modified NN Accuracy | 0.78 | 0.78 | 0.86 | 0.72 | 0.85 | 0.80 | 0.87 | 0.89 |**0.78** | **0.9** | **0.76** |
> > >
> > >
> > > We included all concluded experiments in our updated paper.

---

### Author Response · Authors · 2021-11-11
**Summary and Thanks for Reviewer Comments**

We would like to thank our reviewers for all the constructive and inspiring comments. Thank you, for giving back to the scientific community, especially during difficult times worldwide. First, we would like to very much appreciate the positive assessment of our work:

**Positive points**

**P1: novelty**
“*an interesting piloting idea*”, “*game-changing direction for the future*” and “*may open up a new era of data-driven machine learning in a long run.*” (R0-new); “*human-to-NN path shows novelty*” (R1); “*interesting idea” and “does a good job*” (R3); “*interesting direction to explore*” (R2).
**P2: methodology**
“*A new framework ... to make use of the explanation to allow human intervene the neural net models*” (R0-new); “*Sound methodology*” (R3); “*proposes a sensible approach*” (R2); “*Modifying nodes (adding or removing concepts) are intuitive*” (R1);
**P3: experiments**
“*good experimental results*” (R1); “*evaluation results are interesting and promising*” and “*empirical evaluation is rather insightful for image processing*” (R2); “*results show that this method is useful*” (R3)

For the concerns and questions, we are responding to each reviewer respectively. Thanks!

---

### Author Response · Authors · 2021-11-22
**Conducted more experiments and Updated paper: Experiments, Related work, Discussion, Appendix.**

We have added additional experiments in the paper. In particular,  we ran a Larger experiments of node and edge modification and a Larger experiments of Zero-Shot learning: Here are the details:

(1) **Larger experiments of node and edge modification**: We conducted a larger experiment with ImageNet dataset images. In addition to the original 20 classes, we randomly selected another 100 classes and constructed a 120 classes dataset. For each class, we randomly selected 300 images for training and 200 images for testing. The dataset contains a total of 60000 images. After training, human users chose 16 classes of interest, and used the NN-to-Human path to visualize the reasoning logic (c-SCG) of NN, while the remaining classes were not modified. Then the human users analyzed the c-SCG and modified both nodes (visual concepts) and edges (Concept relationships), one human intervention per class.  The performance of modified classes is shown in the table below:


|   Category        | american_egret| black_stork| spoonbill|white_stork|cheetah|leopard|lion|tiger|
|:----| :----:| :----: | :----: | :----:| :----: | :----: | :----:| :----: |
|Original NN Acc. | 0.51 | 0.31 | 0.70 | 0.88 | 0.48 | 0.85 | 0.33 | 0.50 |
|Modified NN Acc.| 0.58 | 0.40 | 0.75 | 0.92 | 0.57 | 0.89 | 0.41 | 0.54 |

|   Category        | ambulance| beach_wagon| cab|fire_engine|jeep|limousine|recreation_vehicle|school_bus|
|:----| :----:| :----: | :----: | :----:| :----: | :----: | :----:| :----: |
|Original NN Acc. | 0.52 | 0.45 | 0.38 | 0.59 | 0.63 | 0.33 | 0.66 | 0.47 |
|Modified NN Acc.| 0.57 | 0.54 | 0.44 | 0.63 | 0.70 | 0.40 | 0.75 | 0.59 |

For those 104 unmodified classes, their accuracy before and after distillation is consistent (with difference <0.03), their mean accuracy are: (42.6% before distillation and 42.9% after distillation)

(2) **Larger experiments of Zero-Shot learning**: We conducted a larger experiment with OBJECT dataset: We chose 11 objects from OBJECT dataset. 8 of them are seen classes (class A to H) and each class contains 300 training images and 216 testing images. 3 of them are unseen novel classes (class I, J, K) and each unseen class contains only 216 test images. The dataset contains a total of 4776 images.

The table below shows the results:


|   Category        | A| B| C|D|E|F|G|H|I|J|K|
|:----| :----:| :----: | :----: | :----:| :----: | :----: | :----:| :----: |  :----: | :----:| :----: |
|Original NN Accuracy | 0.8 | 0.79 | 0.9 | 0.74 | 0.87 | 0.78 | 0.9 | 0.9 |0 | 0 | 0 |
|Modified NN Accuracy | 0.78 | 0.78 | 0.86 | 0.72 | 0.85 | 0.80 | 0.87 | 0.89 |**0.78** | **0.9** | **0.76** |



We included all concluded experiments in our updated paper. We will pin-point the updated paper changes:

1. (Related work) Adding more related work discussion of Subsection "Human-AI Interaction" (page 2) and Subsection  "Zero-shot learning" (page3).

2. (New Experiements-1) Add larger experiments of node and edge modification into Appendix Section D (in page 17 and 18)

3. (New Experiements-2) Update the larger experiments of Zero-Shot learning in main paper Section 4.2  (page 8 and 9).

4. (Discussion) Add the discussion of “Comparing our zero-shot learning setting with mainstream zero-shot learning settings” at the end of page 9.

---

### Decision · Program_Chairs · 2022-01-20

**Decision:**

Reject

**Comment:**

The paper provides a way for explaining the reasoning of a neural network to humans in the form of a class-specific structural concept graph (c-SCG). The c-SCG can be modified by humans. The modified c-SCG can be incorporated in training a new student model. Experiments show that the new model performs better on classes that their corresponding c-SCG have been modified. While all the reviewers agree that the paper puts forth an interesting idea, some concerns have been raised by reviewers about the scale of experiments and the lack of theoretical guarantee on the fidelity of the SCG. The authors have added two large scale experiments which confirm their previous results as part of their rebuttal. This paper is borderline and needs to be discussed further.